# Ultra-large virtual screening unveils potent agonists of the neuromodulatory orphan receptor GPR139

Israel Cabeza de Vaca [1,11], Boris Trapkov [2,11], Ling Shen [3,11], Duy Duc Vo[1], Xiaoqun Zhang [4], Yunting Yang[4], Mitra Pezeshki [2], Xuehan Zhang[3,5], Frida Bällgren[6], Aljona Saleh[6], Andrii V. Tarnovskiy[7], Dmytro S. Radchenko [7], Yurii S. Moroz [7,8,9], Hans Bräuner-Osborne[2], Per Svenningsson [4], Jan Kihlberg [10], Zhi-Jie Liu [3,5] ✉, Alexander Sebastian Hauser [2] ✉ & Jens Carlsson [1] ✉

The orphan G protein-coupled receptor (GPCR) GPR139 attracts interest as a target for neuropsychiatric disorders. Whereas the physiological functions of GPR139 remain elusive, a high-resolution receptor structure is now available. To assess whether structural information enables ligand discovery, we computationally dock 235 million compounds to the GPR139 binding site. Of 68 top-ranked compounds evaluated experimentally, five are full agonists with potencies ranging from 160 nM to 3.6 μM. Structure-guided optimization identifies one of the most potent GPR139 agonists, and a cryo-EM structure of the receptor-ligand complex confirms the predicted binding mode. Functional characterization provides insights into GPR139 signalling, and one agonist elicits behavioural effects in mice. We also explore the potential to replace experimental structure determination with the deep-learning method Alpha-Fold3, revealing a limited capability of artificial intelligence to model receptor-ligand interactions for understudied GPCRs. The results demonstrate how high-resolution GPCR structures combined with large-library docking can accelerate drug discovery.

Identification of small-molecule ligands of proteins enables drug discovery and remains a cornerstone for development of therapeutic treatments. G protein-coupled receptors (GPCRs) have garnered significant attention due to their pivotal roles in physiological processes. Despite that GPCRs constitute only 15% of the druggable genes in the human genome, this large protein family is the target for more than 34% of approved drugs[1]. Breakthroughs in crystallography and cryo-EM have enabled structure determination for a large number of GPCRs, revealing the molecular basis of ligand binding to drug targets[2]. Furthermore, the recent development of accurate machine learning methods for protein structure prediction has the potential to further increase the structural coverage in the GPCR family[3–5]. Structure-based

[1]Science for Life Laboratory, Department of Cell and Molecular Biology, Uppsala University, Uppsala, Sweden. [2]Department of Drug Design and Pharmacology, University of Copenhagen, Copenhagen, Denmark. [3]iHuman Institute, ShanghaiTech University, Shanghai, China. [4]Neuro Svenningsson, Department of Clinical Neuroscience, Karolinska Institute, Stockholm, Sweden. [5]School of Life Science and Technology, ShanghaiTech University, Shanghai, China. [6]Department of Pharmacy, SciLifeLab Drug Discovery and Development, Uppsala University, Uppsala, Sweden. [7]Enamine Ltd., Kyiv, Ukraine. [8]Taras Shevchenko National University of Kyiv, Kyiv, Ukraine. [9]Chemspace LLC, Kyiv, Ukraine. [10]Department of Chemistry-BMC, Uppsala University, Uppsala, Sweden. [11]These authors contributed equally: Israel Cabeza de Vaca, Boris Trapkov, Ling Shen. ✉e-mail: liuzhj@shanghaitech.edu.cn; alexander.hauser@sund.ku.dk; jens.carlsson@icm.uu.se

design has proven to be an efficient strategy for discovering GPCR ligands and can now be applied to accelerate drug discovery for a large number of therapeutic targets[6,7].

Despite the major interest in the physiological roles of GPCRs, the endogenous ligand and function of many members within this family remain unknown. In fact, more than 100 non-olfactory GPCRs are still classified as orphans, i.e., the endogenous activator has not been identified with certainty[8]. Orphan GPCRs are of great interest for their untapped potential as drug targets and are relevant for a wide range of diseases, including many neuropsychiatric and neurodegenerative conditions[9]. In addition to pairing a receptor with its endogenous ligand, characterization of an orphan is dependent on access to pharmacological tool compounds that can be used to interrogate signaling via different pathways and probe the roles of a receptor in vivo[10,11]. The function of many orphan GPCRs has been deduced based on homology to other receptors, and ligands were identified from libraries of endogenous compounds[12]. Therapeutic exploitation has also been achieved without deorphanization. For example, the agents Lodoxamide (treating allergic keratoconjunctivitis) and Cromolyn (treating idiopathic pulmonary fibrosis and chronic cough), which target the immunomodulatory receptor GPR35, have progressed to clinical trials[13]. However, the rate of receptor deorphanization has decreased in recent years[14]. High-throughput screens often fail to identify starting points for probe development, and many reported ligands of orphans are difficult to reproduce[15]. For these reasons, novel approaches to identify chemical probes are needed to exploit the therapeutic potential of understudied GPCRs[13].

The orphan class A receptor GPR139 was discovered in 2005 and has emerged as a promising drug target[16,17]. Although the endogenous agonist of GPR139 has not been established with certainty, aromatic amino acids (e.g., L-Tryptophan and L-Phenylalanine)[18,19], peptides related to α-melanocyte-stimulating hormone (α-MSH)[20], and dynorphin[21] were demonstrated to activate the receptor. GPR139 activates multiple types of G proteins, and signalling via $G_{q/11}$ is regarded as the primary pathway[18,19,22–30]. The receptor is highly conserved between species, suggesting physiological importance[16], and a large number of studies support that GPR139 has a neuromodulatory function[28,31–34]. The receptor is exclusively expressed in the central nervous system (CNS) within areas involved in movement control, cognition, emotion, behavior, and reward[19,35–38]. The expression of GPR139 is particularly high in the habenula, a neuronal structure implicated in the pathology of psychiatric disorders such as schizophrenia, depression, and attention-deficit/hyperactivity disorder (ADHD)[39–44]. In vivo data showed that GPR139 knockout mice develop neuropsychiatric manifestations with schizophrenia-like symptomatology[31]. Furthermore, genetic variations in the GPR139 locus have been linked to schizophrenia and attention deficit hyperactivity disorder (ADHD)[45,46]. These observations have led to substantial interest in the therapeutic potential of GPR139[22–24] and small-molecule agonists for the treatment of schizophrenia[22]. One of these compounds (**TAK-041**) has been evaluated in clinical trials for the treatment of anhedonia in patients with schizophrenia and major depressive disorder. Although **TAK-041** did not meet the primary endpoints in phase II trials due to lack of efficacy[47,48], GPR139 remains an intriguing orphan receptor with unknown physiological roles and continues to be an interesting drug target for brain disorders[19].

In this work, we explored an integrative approach combining computational and experimental techniques to discover GPR139 ligands. The recent determination of cryo-EM structures of GPR139 in complex with an agonist[49] enabled us to perform a molecular docking screen of a library containing several hundred million molecules. Experimental evaluation of top-ranked compounds identified diverse GPR139 agonists, and the most potent of these was further optimized by utilizing make-on-demand libraries containing billions of compounds. Exploration of structure-activity relationships led to the

discovery of agonists with in vivo behavioral effects, and a cryo-EM structure of the most potent compound bound to the receptor was determined. Characterization of this lead compound in pharmacological assays measuring G protein activation and arrestin recruitment provided insights into the signalling pathways stimulated by GPR139. We also assessed whether receptor-ligand complex structures generated by machine learning methods could serve as effective alternatives to experimentally determined structures for virtual screening targeting understudied GPCRs.

## Results

### Docking screen for GPR139 ligands
A cryo-EM structure of GPR139 bound to the synthetic agonist **JNJ-63553054** was used to identify a ligand binding site[49]. This structure revealed that **JNJ-63553054** binds in a deeply buried pocket that overlaps with the orthosteric site of other class A GPCRs (Fig. 1a). However, the shape and composition of the binding site differ from other GPCRs, and access to this detailed structural information provided a starting point for rational ligand design.

To explore if the GPR139 structure could enable ligand discovery, a chemical library of 235 million lead-like compounds was docked to the binding site. The ultra-large chemical library consisted of diverse compounds with physicochemical properties ideal for hit discovery. The vast majority of these compounds originated from make-on-demand libraries, including numerous scaffolds that had never been synthesized previously (Fig. 1b)[50]. Thousands of compound orientations were sampled in the binding site using DOCK3.7[51], which resulted in the scoring of >200 trillion complexes and required a CPU time equivalent to 6 years on a single core. The top-scoring 300,000 compounds (corresponding to 0.12% of the total library) were then extracted, and molecules with high similarity to previously identified GPR139 ligands and structural motifs known to cause assay interference were removed. The remaining compounds were clustered based on topological similarity to identify a chemically diverse set of candidates for experimental evaluation. The binding modes of 1500 cluster centers were visually inspected. In the compound selection step, we identified compounds forming the interactions similar to **JNJ-63553054** and considered energy contributions that are poorly described by the docking scoring function, as described previously[52,53]. Finally, a set of 68 compounds were selected for in vitro pharmacological characterization and custom synthesized (Compounds **1-68**, Supplementary Table 1).

### Pharmacological assays and discovery of GPR139 agonists
The activity of each predicted ligand at the human GPR139 was first evaluated in functional assays measuring activation of the canonical $G_{q/11}$ protein signaling pathway, which leads to intracellular calcium mobilization. The primary screen was performed at a compound concentration of 10 μM using a calcium mobilization assay and a CHO-K1 cell line stably expressing the human GPR139[24]. Out of the 68 compounds tested for both agonist and antagonist activity at GPR139, 14 compounds (**1–14**) were found to have agonist activity as they stimulated a calcium mobilization response larger than 25% relative to the response observed for the reference **Lundbeck Cmp 1a** (Fig. 1c and Supplementary Table 2). None of the compounds had antagonist activity, as those that inhibited the response of **Lundbeck Cmp 1a** in the antagonist mode screen also produced a response in the agonist mode screen (Supplementary Fig. 1). The 14 compounds were counterscreened at the human muscarinic acetylcholine $M_1$ receptor ($M_1R$) in the same assay and background cell line, which served as a control for potential non-specific interactions and assay interference. None of the compounds stimulated intracellular calcium mobilization response via $M_1R$ at 10 μM, demonstrating that the agonists were GPR139 specific (Supplementary Fig. 2). The 14 hits from the primary screen (compounds **1–14**) were further evaluated at three concentrations (0.1, 1

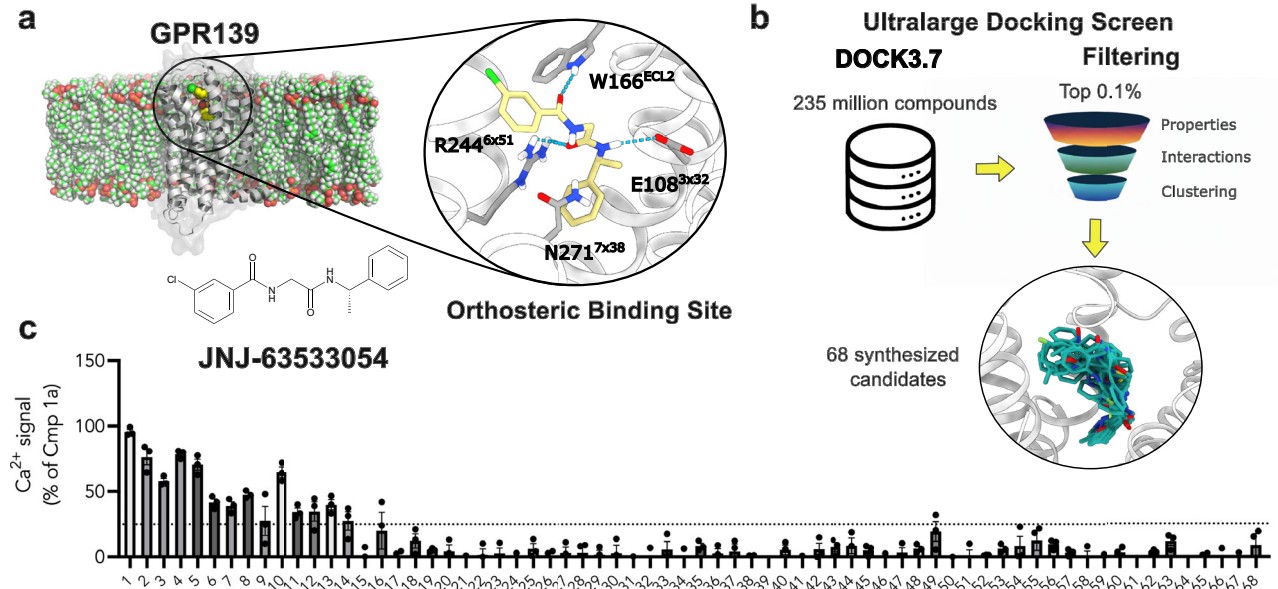

**Fig. 1 | Overview of the virtual screen for GPR139 ligands. a** A structure of the active GPR139 bound to the reference compound **JNJ-63533054** was used in the ultra-large docking screen. **b** A library of 235 million lead-like compounds from the ZINC database was docked to the orthosteric site using DOCK3.7, and a set of 68 top-ranked compounds were selected for synthesis. **c** Primary screening of compounds at $10\,\mu M$ in $Ca^{2+}$ mobilization assay. A threshold of 25% $Ca^{2+}$ response was used to select compounds for further evaluation. Data represent mean ± SEM of at least three independent experiments performed in triplicates and are normalized to buffer (0%) and $10\,\mu M$ of the control (**Lundbeck Cmp 1a**, 100%).

and $10\,\mu M$) (Supplementary Fig. 3). The five most potent agonists (compounds **1–5**) were then characterized with full concentration-response curves. Several of the other nine compounds (e.g., **6, 7** and **10**) showed concentration-dependent activation of GPR139, but their activity was too low to reliably determine $EC_{50}$ values (Supplementary Fig. 3).

Compounds **1–5** showed $EC_{50}$ values ranging from 160 nM to $3.6\,\mu M$ and the $E_{max}$ values spanned 70–100% relative to that of **Lundbeck Cmp 1a** in the calcium mobilization assay (Fig. 2a, b and Supplementary Table 2). Notably, the $EC_{50}$ values of the most potent compounds (**1** and **2**, $EC_{50} = 160$ and 320 nM, respectively) were comparable to the reference compound **Lundbeck Cmp 1a** ($EC_{50} = 200$ nM). We further characterized compounds **1–5** in an orthogonal $G_{q/11}$ signaling assay measuring inositol monophosphate ($IP_1$) accumulation. The compounds stimulated $IP_1$ accumulation with an activity that was consistent with the calcium mobilization experiments (Fig. 2b, c and Supplementary Table 2). The agonist potencies were 3–20-fold lower in the $IP_1$ assay ($EC_{50} = 500$ nM $- 50\,\mu M$), and compounds **3–5** did not saturate the response at the highest tested concentration. This result could be attributed to the larger signal amplification or the spare receptor effect for stimulation of intracellular calcium mobilization relative to $IP_1$ accumulation. However, the rank order in the potency of compounds **1–5** between calcium mobilization and $IP_1$ accumulation assays was maintained. Whereas compounds **1–4** were obtained as racemic mixtures, compound **5** was initially tested as the R-enantiomer ($EC_{50} = 3.6\,\mu M$) due to limited commercial availability. The docking calculations favored the S-enantiomer, and subsequent custom synthesis and evaluation of this compound confirmed that this configuration was more potent in the calcium mobilization assay ($EC_{50} = 540$ nM, $E_{max} = 122\%$).

The identified GPCR139 ligands represented diverse scaffolds and were predicted to capture the key interactions of the reference agonist **JNJ-63533054** (Fig. 2d). **JNJ-63533054** is anchored by phenyl groups in two distinct subpockets of the binding site. The two amide groups connecting these aromatic rings form hydrogen bonds to residues $E108^{3x32}$, $R244^{6x51}$ and $N271^{7x38}$ (superscripts represent generic residue numbering)[54]. Similar to **JNJ-63533054**, all the identified ligands

positioned an aromatic ring in the first subpocket (phenyl or oxadiazole) and amide (**1, 4** and **5**) or urea (**2** and **3**) moieties were predicted to form hydrogen bonds to $E108^{3x32}$ and $R244^{6x51}$. The compounds extended into the second subpocket with more diverse groups. For example, the heterocyclic rings of compounds **1** and **5** formed hydrogen bonds to $W166^{ECL2}$, which were not observed in the cryo-EM structure of **JNJ-63533054**. To assess the chemical similarity of compounds **1–5** to previously described scaffolds, we calculated the maximal ECFP4-based Tanimoto similarity coefficients ($T_c$) to all GPR139 ligands available in the ChEMBL database ($EC_{50} < 10\,\mu M$)[55] (Supplementary Table 3). Whereas compound **5** represented a previously identified scaffold ($T_c = 0.49$), the other four discovered agonists (**1–4**) exhibited lower similarity to known ligands ($T_c < 0.40$). In particular, compounds **2-4** were topologically distinct ($T_c < 0.30$) from all known GPR139 ligands in the ChEMBL database. Interestingly, substructure searches for the core of the most potent compound (**1**, $EC_{50} = 160$ nM, Fig. 2a and 3a) in the ChEMBL database (~2.5 million compounds) yielded no reported bioactivity data. As the scaffold appeared to be unexplored in drug discovery, we prioritized compound **1** for structure-based optimization.

## Structure-guided agonist optimization

Given the promising pharmacology of compound **1**, this agonist was chosen for further exploration of structure-activity relationships. The compound optimization was guided by molecular docking calculations using the GPR139 cryo-EM structure. Two iterations of hit optimization were carried out, which involved searching for analogs among billions of commercial make-on-demand compounds, molecular docking calculations, synthesis, and testing of designed agonists. In total, a set of 7574 analogs of compound **1** was identified, and 44 of these were selected for synthesis and experimental evaluation (Supplementary Table 1).

The 44 analogs (compounds **1.1–1.44**) primarily explored alternatives to the thiophene ring in compound **1**. All analogs, with the exception of compound **1.36**, activated GPR139, and none of the compounds showed antagonist activity in the calcium mobilization assay at a concentration of $10\,\mu M$ (Supplementary Fig. 4). We also

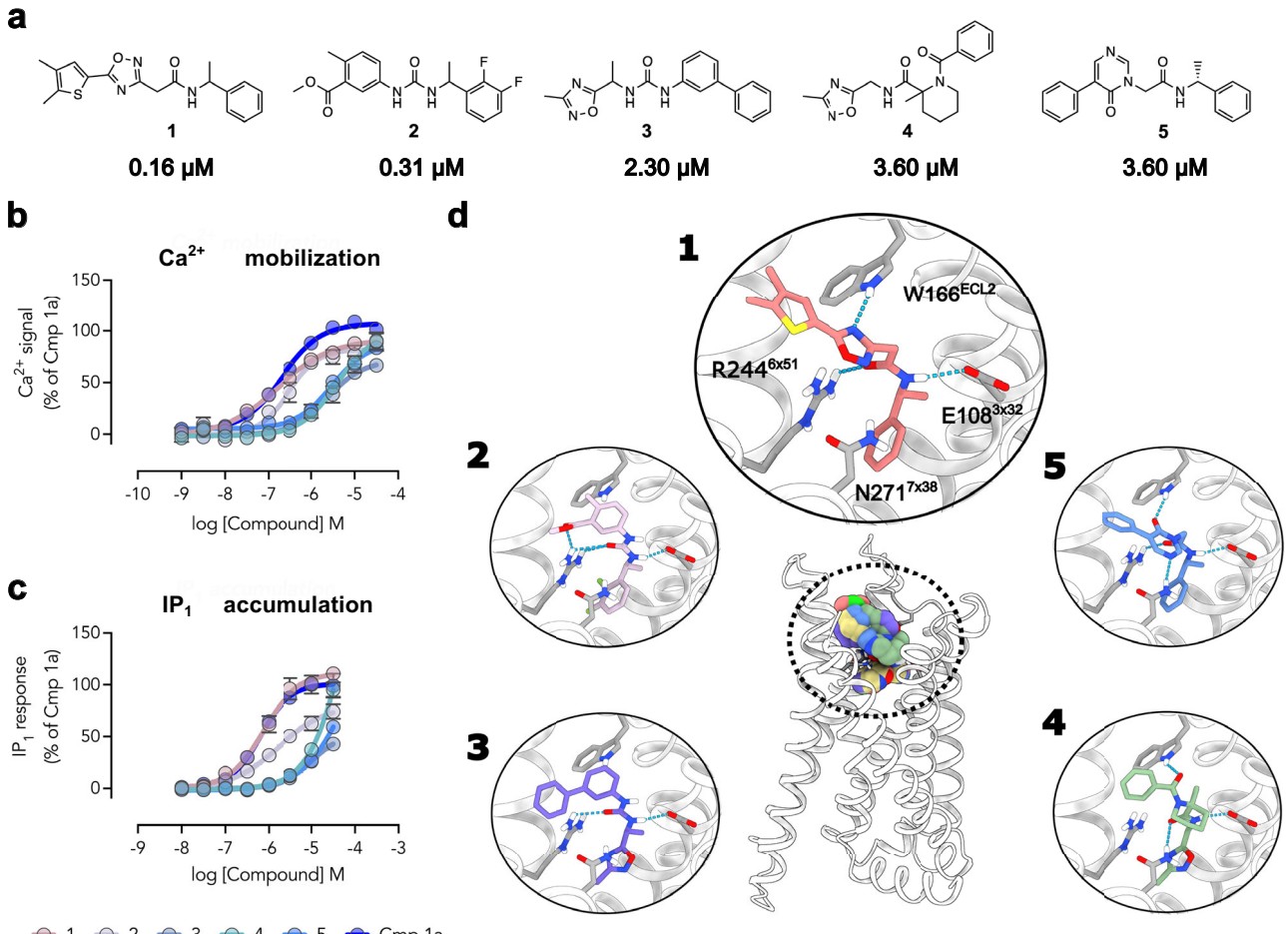

**Fig. 2 | Five GPR139 agonists discovered by the virtual screen. a** Compounds **1**–**5** were pharmacologically characterized for their ability to **b** stimulate intracellular Ca²⁺ mobilization and **c** inositol monophosphate (IP₁) accumulation. Data represent mean ± SEM of at least three independent experiments performed in triplicates and are normalized to buffer (0%) and 10 μM of the control (**Lundbeck Cmp 1a**, 100%).

The potencies (EC₅₀) determined using the Ca²⁺ mobilization assay are shown below each compound structure in (**a**). **d** Predicted binding modes of compounds **1**–**5**. The receptor is depicted as a cartoon with ligands and selected side chains shown in sticks. Hydrogen bonds are indicated using dashed lines. The structures of the predicted complexes are provided in Supplementary Data 1.

controlled for non-specific interactions of these compounds by counter-screening at M₁R. None of the analogs stimulated intracellular calcium mobilization response by M₁R, corroborating their specificity for GPR139 (Supplementary Fig. 5). Concentration-response curves were generated for the 41 analogs that elicited calcium mobilization responses exceeding 50% of the response induced by 10 μM **Lundbeck Cmp 1a** (Supplementary Fig. 6 and Supplementary Table 2). The EC₅₀ values of the compounds ranged from 50 nM to 5 μM (E_max = 66–125%) and revealed structure-activity relationships (Fig. 3a-b).

The parent compound **1** has a 4,5-dimethylated thiophene ring terminally of the molecule structure. The methyl groups in the thiophene ring were not essential for activity, as compounds without these substituents (**1.8**, EC₅₀ = 200 nM) retained high potency relative to parent compound **1** (EC₅₀ = 160 nM). However, a methyl group in position 4 was beneficial, as shown by compound **1.1** (EC₅₀ = 50 nM), which exhibited a 3-fold improved potency. The presence of a methyl group in other positions of the thiophene ring did not influence the potency (**1.4, 1.9,** and **1.11**, EC₅₀ = 100-200 nM). Larger substituents on the thiophene ring were generally less tolerated, decreasing potency from a three-fold reduction to a complete loss of activity (**1.12, 1.21, 1.22, 1.28, 1.36, 1.40,** and **1.41**). Substituting the thiophene ring to a furan ring did not influence the potency (**1.9** and **1.6**). Substituting the thiophene with a more polar thiazole ring was less tolerated and decreased potency by 3- to 25-fold relative to parent compound **1** (**1.15,**

**1.34,** and **1.43**, EC₅₀ = 500–4000 nM). Furthermore, an oxazole ring generally decreased the potency (**1.23, 1.25, 1.29,** and **1.30**, EC₅₀ = 1600–2500 nM), but the activity was maintained for a 3-methylisoxazole ring (compound **1.5**, EC₅₀ = 130 nM). The SAR was further explored by replacing the thiophene ring with a methylated or methoxylated phenyl or pyridine ring. The phenyl ring was well tolerated, but the position of the methyl substituent influenced potency (**1.3, 1.14,** and **1.20**, EC₅₀ = 100–1300 nM). Methyl substituted benzenes showed slightly higher potencies than the methoxy substituents (**1.3** vs **1.10**, and **1.14** vs **1.18**, EC₅₀ = 100 vs 200 nM, and 400 vs 1000 nM). The presence of a methoxy group in position 2 was beneficial (compound **1.2**, EC₅₀ = 50 nM), leading to a 3-fold improved potency compared to compound **1**. The pyridine ring was generally less tolerated than the phenyl and led to a 4- to 16-fold decrease in the potency relative to a similar phenyl substituted compound (**1.13, 1.17, 1.19,** and **1.24** vs **1.3**, EC₅₀ = 400–1600 nM vs 100 nM). Finally, a key feature of compound **1** is the central oxadiazole ring, which was crucial for the activity of our compounds as substitution to a triazole or oxazole ring decreased the potency by 7- and >50-fold (**1.38** and **1.44**, EC₅₀ = 1300 nM and 43% response at 10 μM, respectively) (Fig. 3b). A subset of representative analogs was also evaluated for their ability to stimulate the IP₁ accumulation (Supplementary Fig. 7 and Supplementary Table 2). In agreement with the calcium mobilization assays for the virtual screening hits, the compounds stimulated IP₁

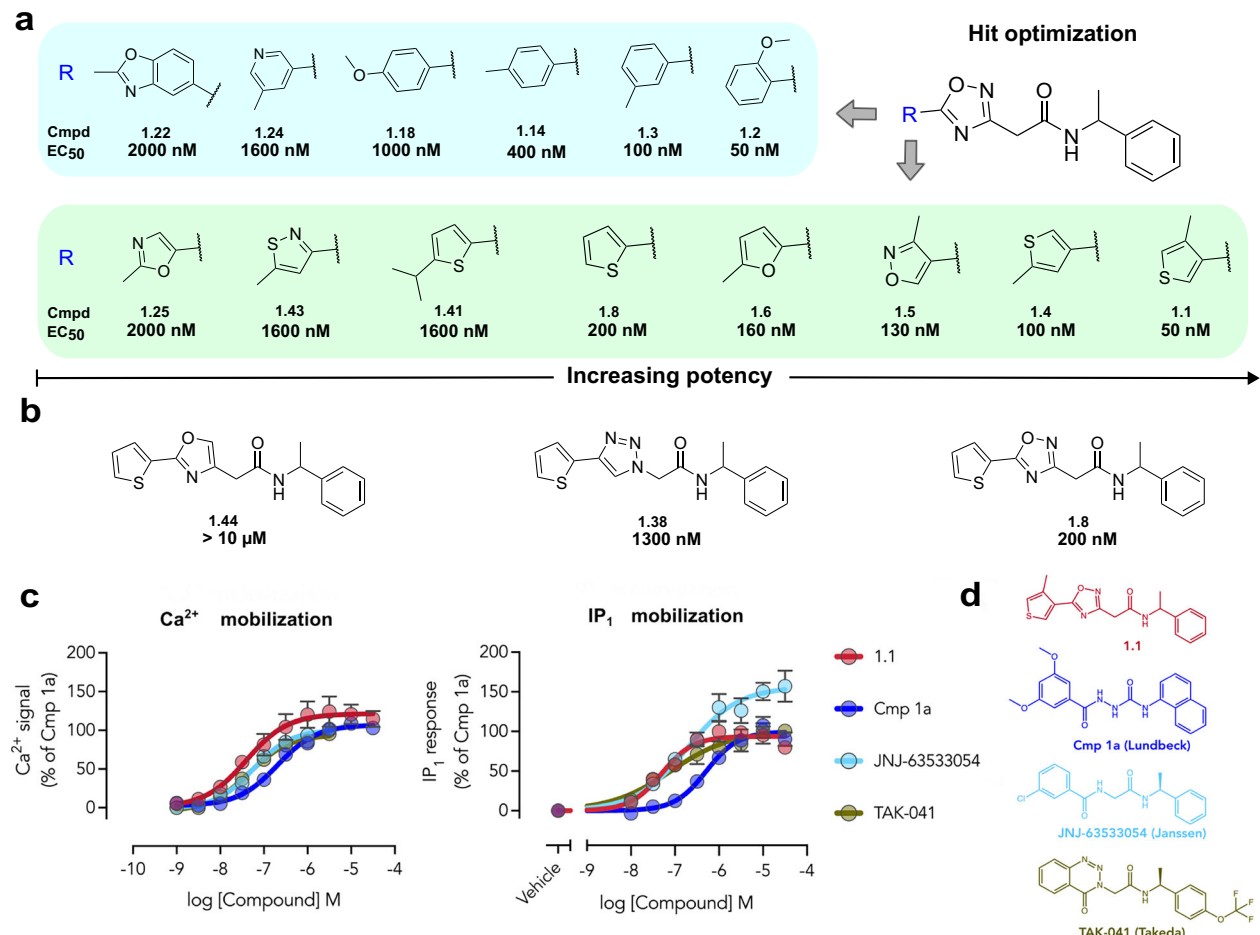

**Fig. 3 | Optimization and structure-activity relationships of GPR139 agonists.**
**a**, **b** Structure of core scaffold (R = aromatic ring) explored in optimization and representative analogs of compound **1** with EC$_{50}$ values from the Ca$^{2+}$ mobilization assay. The blue and green areas in (**a**) show analogs based on six-membered and five-membered ring substituents, respectively. **c** The reference GPR139 agonists **Lundbeck Cmp 1a, JNJ-63533054, TAK-041**, and compound **1.1** were evaluated for their ability to stimulate intracellular Ca$^{2+}$ mobilization and IP$_1$ accumulation. Data represent mean ± SEM of at least three independent experiments performed in triplicates and are normalized to buffer (0%) and 10 µM of the control (**Lundbeck Cmp 1a**, 100%). **d** Chemical structures comparison of the reference agonists and compound **1.1**.

accumulation with slightly lower potencies, and the overall rank order in the potency was maintained. A notable exception was that the most promising compound **1.1** showed equal potencies in calcium mobilization and IP$_1$ accumulation assays (50 nM). We compared compound **1.1** to three reference compounds (**Lundbeck Cmp 1a, JNJ-63533054,** and **TAK-041**) for their ability to stimulate intracellular calcium mobilization and IP$_1$ accumulation. Compound **1.1** was slightly more potent than **Lundbeck Cmp 1a** and **JNJ-63533054**, and showed comparable potency to **TAK-041**, with the most pronounced difference in the IP$_1$ accumulation assay (Fig. 3c, d and Supplementary Table 4).

**Cryo-EM structures confirm the computationally predicted agonist binding mode and reveal limitations of AlphaFold3 models**
To gain further insights into the molecular basis of ligand recognition by GPR139 and assess the accuracy of our predicted binding mode, a structure of the complex with compound **1.1** was determined (PDB accession code: 9M42). The cell expression and purification of GPR139 samples were described previously[49]. The GPR139–mini-G$_{s/q}$ complex was constituted in vitro with purified GPR139, mini-G$_{s/q}$ protein[49,56], and stabilizing Nb35, in the presence of a racemic mixture of compound **1.1**. After a few rounds of optimization in sample preparation and data collection, the cryo-EM complex structure of

GPR139–compound **1.1**–mini-G$_{s/q}$–Nb35 was determined at an overall resolution range of 3.2 Å (Fig. 4a–c and Supplementary Fig. 8). Of the two enantiomers, the S-form of compound **1.1** (**1.1(S)**) showed a clearer and more consistent fit to the ligand density than the R-form (Fig. 4a–c). The experimentally determined binding mode of compound **1.1(S)** agreed well with the docking model (ligand root-mean-square deviation (RMSD) of 2.9 Å), which correctly identified the regions occupied by the phenethylamide and 1,2,4-oxadiazol moieties (Fig. 4d).

Comparative structural analysis between the compound **1.1(S)**-bound and **JNJ-63533054**-bound GPR139 structures reveals only minor differences, with a receptor Cα RMSD of 0.98 Å. Notably, compound **1.1(S)** appears to induce an outward expansion of the extracellular portion of the binding pocket (Supplementary Fig. 9a). To quantify these conformational changes, we measured the displacement of representative residue Cα atoms from each transmembrane helix (TM) between the superimposed compound **1.1(S)**- and **JNJ-63533054**-bound GPR139 structures. Specifically, the measured distances for the selected residues at the extracellular ends of the helices ranged from 0.5 to 2.1 Å (1.7 Å for F27[1x33] of TM1, 2.1 Å for D89[2x66] of TM2, 1.1 Å for K102[3x26] of TM3, 0.5 Å for L158[4x55] of TM4, 0.8 Å for V182[5x37] of TM5, 1.6 Å for Y250[6x57] of TM6, and 1.6 Å for H264[7x32] of TM7). Comparison of the superimposed complexes also revealed differences between the

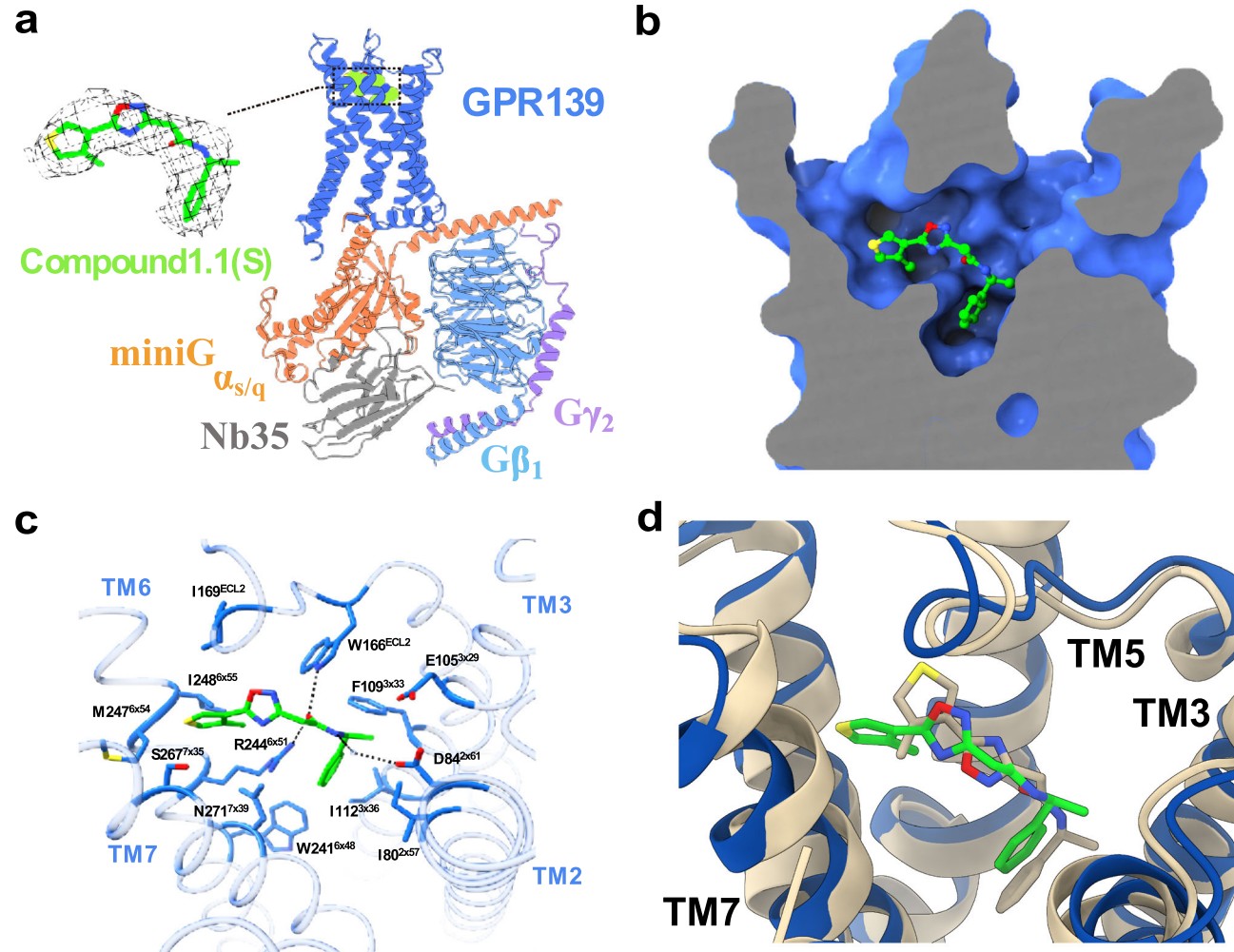

**Fig. 4 | Analysis of the cryo-EM structure of the GPR139–G protein complex bound to compound 1.1(S). a** Cartoon representation of the GPR139–compound **1.1(S)**–miniG$_{s/q}$–Nb35 complex, along with the density map of compound **1.1(S)** (PDB accession code: 9M42). **b, c** Pose and key residues involved in the binding of compound **1.1(S)** (green sticks) within the GPR139–miniG$_{s/q}$ complex (dark blue). **d** Comparison of the predicted binding mode of compound **1.1** (brown) and the structure determined by cryo-EM (blue receptor, green ligand). The receptor is depicted as a cartoon and ligands are shown as sticks.

binding modes and interactions of compound **1.1(S)** and **JNJ-63533054** (Fig. 4c and Supplementary Fig. 9b). In the structure of compound **1.1(S)**, the phenethylamide is shifted toward the extracellular side of the binding pocket compared to the position of the same group in **JNJ-63533054**. The phenyl group can maintain a cation-π interaction with residue R244$^{6×51}$ in both complexes, but there are distinct differences in the interactions with the cluster of negatively charged residues in TM2 and TM3. Whereas the amide nitrogen of **JNJ-63533054** is located near E108$^{3×32}$, compound **1.1(S)** instead forms a water-mediated hydrogen bond with D84$^{2×61}$. In addition, the amide carbonyl of **JNJ-63533054** forms a hydrogen bond with the side chain of R244$^{6×51}$, whereas, the same group in compound **1.1(S)** can form polar contacts with both R244$^{6×51}$ and W166$^{ECL2}$. The shift in ligand binding mode may be due to the 1,2,4-oxadiazole group in compound **1.1(S)** replacing the second amide in **JNJ-63533054**, which is engaged in a polar interaction network involving residues N271$^{7×38}$ and W166$^{ECL2}$. Compared to the amide group in **JNJ-63533054**, the less polar 1,2,4-oxadiazole moiety is shifted towards TM6 and instead occupies the position of the chlorobenzyl group in **JNJ-63533054**. The thiophene group of compound **1.1(S)** extends into a pocket formed by TM6 and TM7 that is not occupied by **JNJ-63533054**, pushing these helices outward and contributing to the expansion of the binding site.

Advances in machine learning have revolutionized protein structure prediction, and the AlphaFold2 algorithm often generates models

of receptors with near-experimental accuracy. The recently released AlphaFold3 (AF3) can also model receptors in complex with small molecules[5]. Although AF3 is two orders of magnitude slower than DOCK3.7 and hence lacks the computational efficiency to screen large chemical libraries, the method has been demonstrated to predict ligand binding modes with impressive accuracy. In the case of GPR139, access to detailed information regarding the location of the ligand and key interactions with the receptor was crucial for enabling the docking screen. This raises the question of whether AF3 could provide structures suitable for structure-based virtual screening, potentially replacing resource-intensive cryo-EM and X-ray crystallography experiments. Encouragingly, we found that AF3 could predict the binding mode of the agonist **1.1(S)** in the GPR139 binding site with good accuracy (ligand RMSD = 2.7 Å, captured receptor-ligand contacts, LC = 32%, Supplementary Fig. 9c, d), a result that was comparable to that obtained using molecular docking (ligand RMSD = 2.9 Å, LC = 10%, Fig. 4d). However, it should be noted that AF3 training included structures of GPR139, e.g., the one used in the docking screen. Predicting complexes for understudied GPCRs that were not part of the training set provides a more relevant test set to benchmark performance. To evaluate the ability of AF3 to identify the ligand binding site, we extended the calculations to five recently determined small molecule complexes of orphan GPCRs that were not part of the training set (Supplementary Table 5 and Supplementary Fig. 10–12). AF3

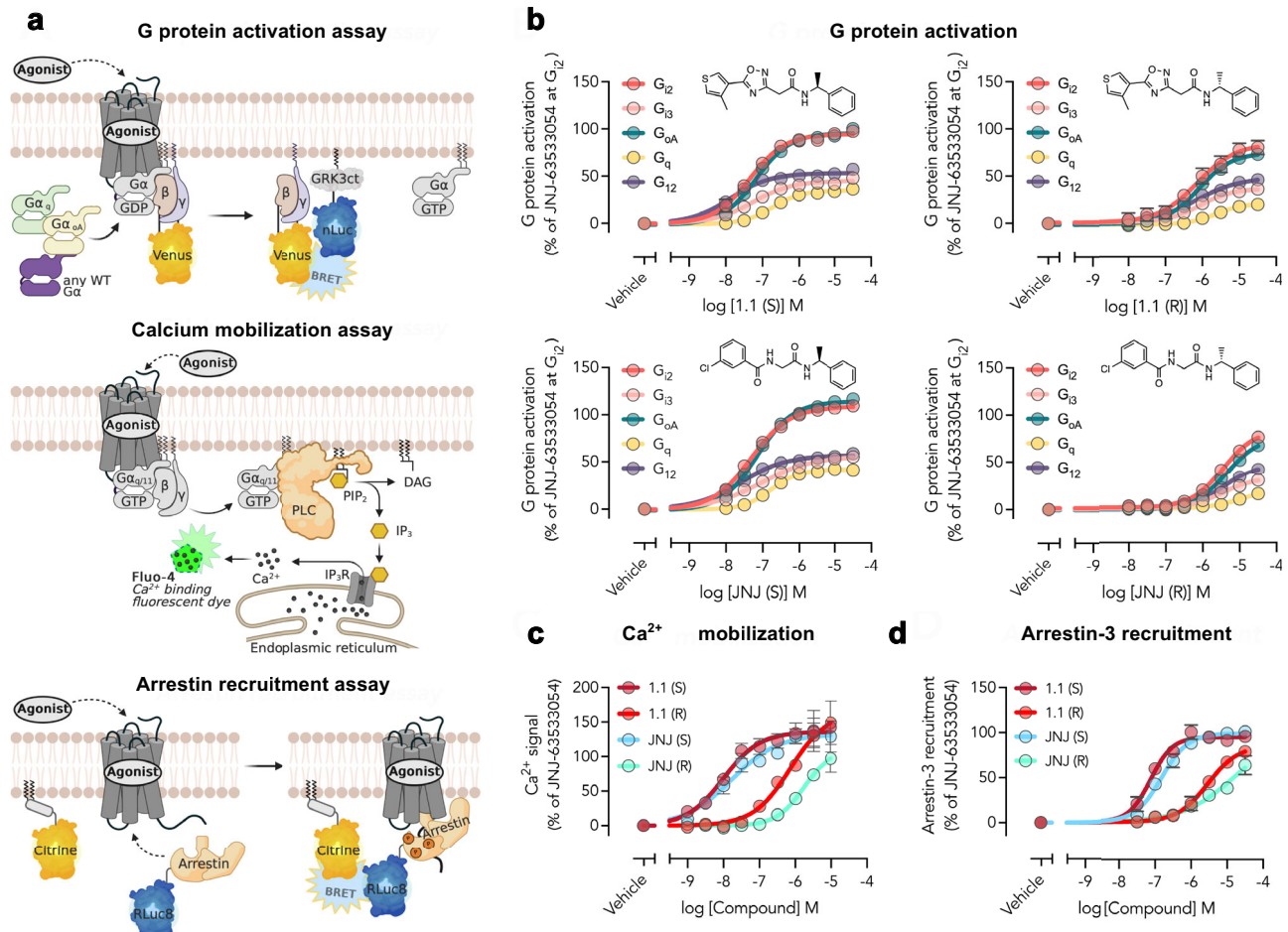

**Fig. 5 | Characterization of the stereoisomers of compound 1.1 in signaling assays.** The ability of the S- and R-enantiomers of compound **1.1** and **JNJ-63533054** (**JNJ**) to stimulate GPR139-mediated signaling pathways was evaluated. **a** Schematic representation of the signaling pathway assays. BioRender created graphics. Concentration-response curves for the **b** activation of $G_{i2}$, $G_{i3}$, $G_{oA}$, $G_q$, and $G_{12}$ proteins, **c** stimulation of intracellular $Ca^{2+}$ mobilization, and **d** stimulation of arrestin-3 recruitment by transiently expressed GPR139. Data represent mean ± SEM of at least three independent experiments performed in triplicates and are normalized to buffer (0%) and 32 μM of the reference compound (**JNJ-63533054**, 100%). Created in BioRender. Trapkov, B. (2025) https://BioRender.com/99t6t3c.

predictions only achieved reasonable accuracy for one of the complexes (GPR84: ligand RMSD = 3.4 Å, LC = 21%). For the four other complexes, AF3 was able to identify the correct binding pocket in one case, but with high ligand RMSD (ligand RMSD = 5.7 Å, LC = 2%). In the remaining three cases, the AF3 placed the ligand outside the orthosteric binding side (ligand RMSD = 6.3–20.3 Å, LC = 0–2%) and these models would hence be unsuitable as starting points for docking screens.

### Signalling signature of the potent GPR139 agonist 1.1

We evaluated the signaling pathways activated by the potent agonist **1.1** and **JNJ-63533054** compounds through GPR139. Previous studies of **JNJ-63533054** and **TAK-041** have demonstrated the S-enantiomer to be the most active configuration of chiral GPR139 agonists[22,23]. As all our screens of analogs were performed with racemic mixtures, we synthesized compound **1.1** as pure S- and R-enantiomer (**1.1(S)** and **1.1(R)**, respectively) along with **JNJ-63533054** (**JNJ(S)**)) and its R-enantiomer (**JNJ(R)**). Detailed synthetic procedures are described in the Supplementary Methods. We then characterized these four compounds in assays measuring the activation of $G_{i2}$, $G_{i3}$, $G_{oA}$, $G_q$, and $G_{12}$ proteins, intracellular calcium mobilization, and arrestin-3 recruitment. G protein activation was measured using a BRET-based Gα-specific G protein dissociation assay[57] and arrestin-3 recruitment was measured with an enhanced bystander BRET assay[58] (Fig. 5a).

All four compounds, **1.1(S), 1.1(R), JNJ(S)**, and **JNJ(R)**, stimulated $G_{i2}$, $G_{i3}$, $G_{oA}$, $G_q$ and $G_{12}$ protein activation, intracellular calcium mobilization and arrestin-3 recruitment by GPR139 (Fig. 5b–d). The S-enantiomers exhibited the highest level of activity, which was also consistent with the cryo-EM structure of compound **1.1** bound to GPR139. Compound **1.1(S)** was 13- to 63-fold more potent than **1.1 (R)**, while **JNJ(S)** was 40- to 158-fold more potent than **JNJ(R)** (Supplementary Table 6). Both **1.1(S)** and **JNJ(S)** generally had comparable potency for stimulating the individual signaling pathways at GPR139, although there was a tendency for **1.1(S)** to be slightly more potent than **JNJ(S)** (Supplementary Table 6). The $EC_{50}$ of **1.1(S)** for stimulating the individual signaling pathways at GPR139 ranged from 13 to 200 nM. To the best of our knowledge, compound **1.1(S)** is hence one of the most potent GPR139 agonists published to date. The results are further supported by the calcium mobilization experiments with **1.1(S)** in a stable GPR139-CHOk1 cell line demonstrating an $EC_{50}$ of 8 nM (Supplementary Fig. 13 and Supplementary Table 7). Interestingly, the most potent responses at GPR139 were measured for $G_{12}$ protein activation, in addition to calcium mobilization. $G_{12}$ protein coupling by GPR139 has not previously been reported, and we were able to demonstrate robust coupling by the receptor. Our comprehensive analysis of activated GPR139 signaling pathways hence showed that **1.1(S), 1.1(R), JNJ(S)**, and **JNJ(R)** exhibited comparable signaling profiles at GPR139 (Supplementary Table 6).

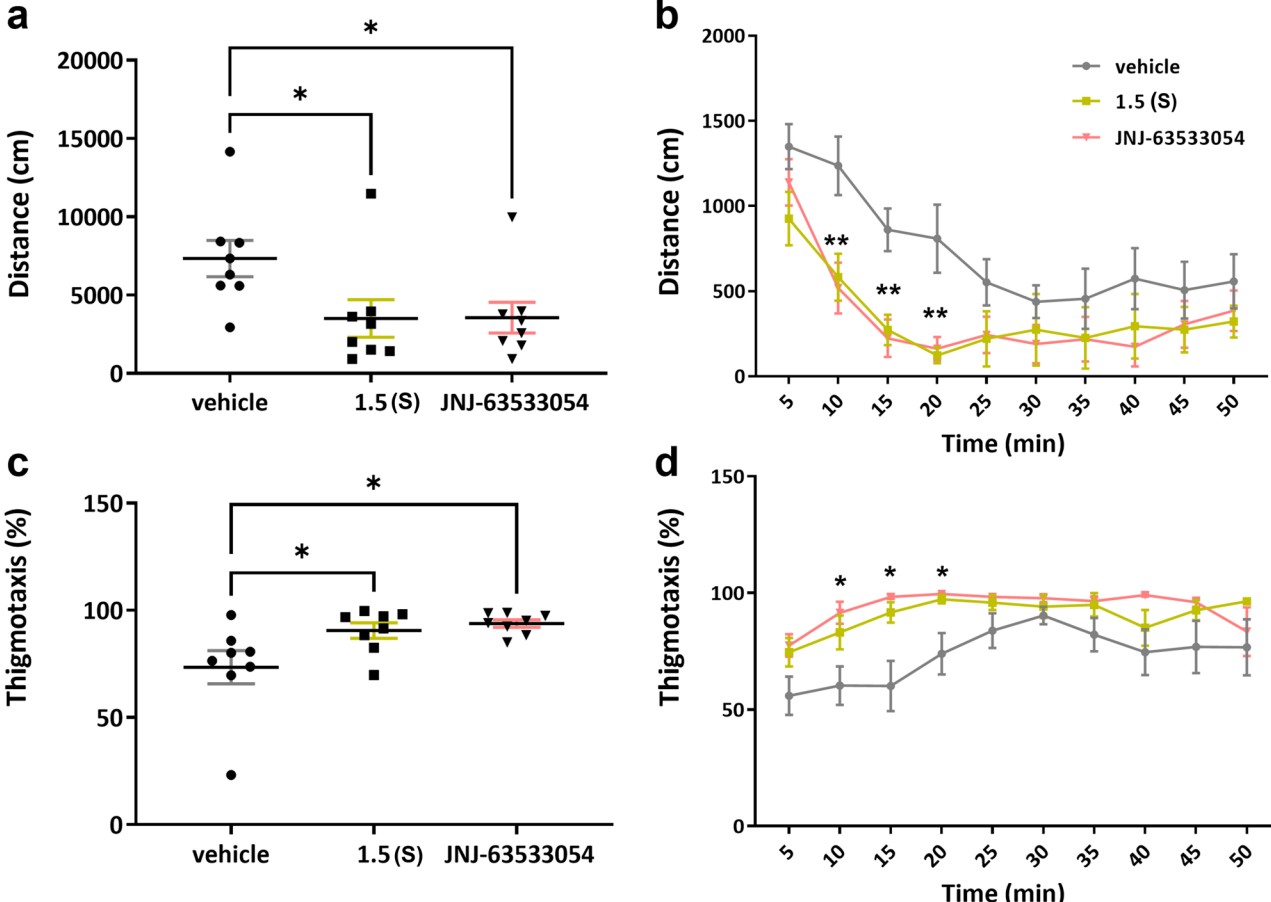

**Fig. 6 | Effects of compound 1.5(S) and JNJ-63533054 on locomotion and thigmotaxis. a** Total distance travelled in 50 mins by mice treated with 10% DMSO + 10% solutol HS-15 + 80% saline ($n = 8$), compound **1.5(S)** ($n = 8$, 30 mg/kg, i.p.), or **JNJ-63533054** ($n = 8$, 30 mg/kg i.p.), both of which were dissolved in 10% DMSO + 10% solutol HS-15 + 80% saline. **b** Time course of the distance travelled by mice treated with 10% DMSO + 10% solutol HS-15 + 80% saline ($n = 8$), compound **1.5(S)** ($n = 8$, 30 mg/kg, i.p.), or **JNJ-63533054** ($n = 8$, 30 mg/kg i.p.), both of which were dissolved in 10% DMSO + 10% solutol HS-15 + 80% saline. **c** Thigmotaxis in 50 min in mice treated with 10% DMSO + 10% solutol HS-15 + 80% saline ($n = 8$), compound **1.5(S)** ($n = 8$, 30 mg/kg, i.p.), or **JNJ-63533054** ($n = 8$, 30 mg/kg i.p.), both of which were dissolved in 10% DMSO + 10% solutol HS-15 + 80% saline and **d** the time course of the thigmotaxis in mice treated with 10% DMSO + 10% solutol HS-15 + 80% saline ($n = 8$), compound **1.5(S)** ($n = 8$, 30 mg/kg, i.p.), or **JNJ-63533054** ($n = 8$, 30 mg/kg i.p.), both of which were dissolved in 10% DMSO + 10% solutol HS-15 + 80% saline. Data represent mean ± S.E.M. *$p < 0.05$, **$p < 0.01$ (**a**: One-way ANOVA, $F_{(2, 21)} = 3.838$ $P = 0.0380$; post hoc test with Dunnett's multiple comparisons test, vehicle vs. 1.5(S), $p = 0.0457$; vehicle vs. **JNJ-63533054**, $p = 0,0485$, **b**: Two-way ANOVA, Time*Treatmen: $F_{(18, 210)} = 0.7424$, $P = 0.7646$; Time: $F_{(9, 210)} = 10,75$, $P < 0.0001$; Treatment: $F_{(2, 210)} = 23,67$, $P < 0.0001$ followed by Tukey's multiple comparisons test; **c**: One-way ANOVA, $F_{(2, 21)} = 4,679$ $P = 0.0209$; post hoc test with Dunnett's multiple comparisons test, vehicle vs. 1.5(S), $p = 0.0476$; vehicle vs. **JNJ-63533054**, $p = 0.0181$; and **d**: Two-way ANOVA, Time*Treatment: $F_{(18, 210)} = 0.8496$, $P = 0.6401$; Time: $F_{(9, 210)} = 4.098$, $P < 0.0001$; Treatment: $F_{(2, 210)} = 29,05$, $P < 0.0001$ followed by Tukey's multiple comparisons test).

## In vivo behavioural activity of potent GPR139 agonists in the open-field test

To further explore the therapeutic potential of the GPR139 agonists, we evaluated in vivo activity of the potent scaffold in mice in the open-field test to assess locomotion and anxiety. As compound **1.1(S)** showed low metabolic stability in the presence of mouse liver microsomes (intrinsic clearance, $CL_{int} = 1369$ µL/min/mg) and moderate kinetic solubility (13 µM), another potent agonist in our series, compound **1.5(S)**, was used in the animal studies ($pEC_{50} = 6.9$, $CL_{int} = 46$ µL/min/mg, and solubility = 24 µM for the racemic form).

To evaluate whether our compounds could penetrate the blood-brain barrier and activate GPR139, we studied the effects of compound **1.5(S)** and reference agonist **JNJ-63533054** in an open field test (30 mg/kg i.p.). A previous study has shown that **JNJ-63533054** induces a reduction in locomotor activity in the first hour after treatment[19]. The mice treated with compound **1.5(S)** showed a significant decrease in total distance travelled in the open field arena, with results similar to those obtained for **JNJ-63533054** (Fig. 6a). The time course showed that the effects were more pronounced at the 10- to 20-min time points

(Fig. 6b). Mice treated with compound **1.5(S)** and **JNJ-63533054** spent more time in the periphery compared to the vehicle group, as indicated by an increase of the thigmotaxis, suggesting increased anxiety-like behaviour (Fig. 6c). This result is in agreement with the reduced thigmotaxis observed in GPR139 knock-mice in previous studies[31]. The time course showed that the effects were more pronounced at the 10- to 20-min time points (Fig. 6d), which was consistent with the decrease in the total distance travelled (Fig. 6b).

One hour after administration of 30 mg/kg i.p., the total concentrations of **1.5(S)** and **JNJ-63533054** were determined in mouse plasma and brain tissue. For **1.5(S)**, the total plasma concentration ($C_{plasma}$) was determined to 8.0 ± 2.2 (3.8–21.1) µM, and the total brain tissue concentration ($C_{brain}$) to 4.8 ± 1.0 (3.2–14.7) µM. For **JNJ-63533054**, $C_{plasma}$ was determined to 20.5 ± 5.3 (12.9–29.1) µM, and $C_{brain}$ to 24.8 ± 6.2 (18.7–36.0) µM. Single-time-point total brain-to-plasma concentration ratios ($K_{p,brain}$)[59] for **1.5(S)** and **JNJ-63533054** were calculated by dividing the total brain concentration by the total plasma concentration. The $K_{p,brain}$ values were 0.67 ± 0.15 for **1.5(S)** and 1.21 ± 0.12 for **JNJ-63533054**. Both tested compounds thus reached

the brain and were present at total concentrations exceeding their in vitro potencies.

## Discussion

The development of chemical probes is crucial for advancing research in pharmacology, enabling characterization of receptor physiology and catalyzing drug discovery. Our efforts to identify ligands of the orphan receptor GPR139 yielded three key results, which demonstrate the potential of using structure-based virtual screens to leverage ultra-large chemical libraries. First, our docking screen of several hundred million compounds against GPR139 identified five agonists, including diverse scaffolds that represent promising starting points for drug discovery. Second, we optimized one of the compounds using structure-guided design, resulting in a highly potent agonist. Determination of a cryo-EM structure of the GPR139-agonist complex confirmed the predicted ligand binding mode, and the compound was extensively characterized in signalling assays, unveiling $G_{12}$ protein coupling by GPR139. Finally, we demonstrated an in vivo effect of an agonist in the open-field test in mice, indicating actions on both locomotion and emotionality.

The large expansion of commercially available chemical space has initiated a paradigm shift in virtual screening[7,60]. Structure-based docking calculations can now search in libraries containing billions of diverse chemical structures, and promising compounds can readily be synthesized and tested experimentally for biological activity. Several recent virtual screening studies have employed this approach to identify potent GPCR ligands. The first screens primarily focused on extensively studied drug targets in the GPCR family, such as adrenergic, dopamine, and serotonin receptors[61–64]. The relatively small, enclosed orthosteric pockets of these receptors, together with well-defined key interactions for binding, facilitate identification of ligands in chemical libraries. Docking to such sites often leads to high hit rates (20-70%) and the receptors bind ligands with high affinities[6]. In contrast, the features of the GPR139 extracellular pocket are more complex in shape and polarity, and the receptor responds only weakly to the amino acids proposed as its endogenous ligands. The more challenging nature of the target likely explains why our screening hit rate (7%)—although orders of magnitude higher than generally observed in high-throughput screening—is lower compared to docking screens targeting aminergic receptors. The docking hit rate is more similar to those obtained for more challenging targets, such as peptide or protein-binding receptors[6], in agreement with the hypothesis that GPR139 is a peptide-activated GPCR[21]. A more relevant comparison is that we achieved a two-fold higher hit rate than a previous virtual screening study targeting GPR139[29]. Although it should be noted that our lead molecules contain a phenethylamide moiety, which is also present in known GPR139 ligands (e.g., **JNJ-63533054**), this series of agonists reached very high potencies. In fact, compound **1.1** is likely one of the most potent GPR139 agonists identified to date[22–24,26,27,29,65], and our study is among the few on GPCRs that validated the predicted binding mode by determining the structure of the complex[62,64]. These results highlight the potential of structure-based screens of ultra-large libraries to accelerate drug discovery. Further optimization of the scaffold represented by compound **1.1** can be guided by the cryo-EM structure and should also consider pharmacokinetic properties and target selectivity. Due to the limited number of analogs available for each scaffold in commercial chemical space, lead optimization may require exploring compounds not present in these libraries[66]. Compound design may also benefit from accounting for binding site flexibility and employing computational methods that predict ligand affinities more accurately than docking scoring functions[67].

Cryo-EM structures of understudied GPCRs are now rapidly emerging, uncovering binding pockets that can be leveraged for drug discovery[68]. Access to accurate information regarding the location and shape of ligand binding sites is essential for the success of docking screens. Due to the low sequence similar to other GPCRs, prediction of the structure and ligand binding site of GPR139 with traditional modeling techniques was challenging. In a community-wide assessment of GPCR structure prediction (GPCR Dock 2021), which included GPR139 in complex with **JNJ-63533054**, only one out of 198 submitted models achieved a ligand RMSD $\leq 2\,\text{Å}$[69]. Our docking screen was enabled by a cryo-EM structure of GPR139 bound to a synthetic agonist[49]. Guided by the structure, we focused on identifying compounds occupying the identified agonist binding pocket, a decision that strongly influenced the results of the screen. All the five identified ligands recapitulated the efficacy of the bound agonist, and the most potent compound even had the same G protein signaling profile as the reference agonist **JNJ-63533054**. These results suggest that a structure of the inactive GPR139 or design of ligands targeting other subpockets may be required to identify antagonists. The benefits of access to experimentally determined structures of receptor-ligand complexes for understudied GPCRs were further highlighted by our evaluation of AF3, a state-of-the-art deep learning method for prediction of biomolecular complexes[5]. Although machine learning-generated receptor models have recently been used successfully in docking screens[64,70], our analysis supports recent studies concluding that deep learning methods appear to lack the accuracy required to model ligands if a near-neighbor case was not present in the training data. Consequently, the predicted ligand binding modes cannot be expected to be reliable for many understudied receptors and binding sites, limiting the utility of these models in virtual screening[71].

The development of drugs to treat brain disorders remains challenging, and neuropsychiatric diseases are particularly difficult. In fact, the recently approved drug KarXT, an $M_1/M_4$ muscarinic receptor agonist, is the first treatment for schizophrenia with a new mechanism of action in several decades[31]. Although the clinical trial of a GPR139 agonist (**TAK-041**) was unsuccessful, this compound had a good safety profile and was well tolerated[72]. The link between GPR139 and the dopamine and opioid neuromodulatory systems[28,31,33], which play central roles in the pathophysiology of neuropsychiatric disorders, positions GPR139 as an intriguing neurological target that warrants further exploration. In addition to schizophrenia, GPR139 modulators are also relevant for other indications, such as pain, neurodegenerative, metabolic, and substance use disorders[17]. To date, only a few in vivo studies for GPR139 have been conducted, and more work is required to understand the function of this highly conserved orphan in health and disease. Adding to the GPR139 biology, our extensive evaluation of signaling pathways resulted in deeper understanding of the receptor's signaling, and we report several findings, including $G_{12}$ protein coupling and arrestin recruitment. $G_{12}$ plays an important role in physiology and is well known for its function in cellular development and growth. This G protein is also involved in the nervous system, with roles in neuronal migration, axonal guidance, the formation of cerebellar and cerebral cortices, and neurotransmitter release[73]. Specifically, $G_{12}$ has been implicated in stress-related neuropsychiatric disorders[74]. Studies in native cells will be required to understand the relevance of GP139 signaling via different G proteins. Our results might benefit future studies aiming to dissect the physiological function of GPR139 and lay a foundation for design of GPR139 ligands with biased signaling profiles. Finally, our study provides a highly potent agonist scaffold exhibiting in vivo activity, which will enable further characterization of GPR139 and its potential as a therapeutic target for neuropsychiatric diseases.

## Methods

### Computational and medicinal chemistry

**Molecular docking.** Molecular docking calculations were performed using a cryo-EM structure of GPR139 bound to **JNJ-63533054**. Based on the cryo-EM density map (PDB accession code 7VUG), **JNJ-63533054** can adopt two distinct poses (pose-1 and pose-2; see Supplementary

Fig. 7b in ref. [49]). The complex with pose-2 was used in the virtual screen and is provided in Supplementary Data 1. Solvent molecules were removed from the structure, and the ligand atoms were used to define the binding site by generating 45 matching spheres. The DOCK3.7 docking software uses a flexible ligand algorithm overlaying the rigid segments of ligands with precomputed conformations onto these matched spheres[51]. The GPR139 structure was protonated using the REDUCE software[75], and the AMBER united atom force field[76] was used to obtain the partial charges of the atoms. All histidine protonation states were assigned automatically to Nε by REDUCE based on the hydrogen bonding networks. E105$^{3x29}$ and D84$^{2x61}$ were neutralized manually based on visual inspection of the binding site. The dipole moments of two residues involved in the recognition of the **JNJ-63533054** reference molecule were increased to favor interactions with these. This approach is a common practice to improve docking performance in DOCK3.7[53]. For the residues W166$^{ECL2}$ and R244$^{6x51}$, the partial atomic charges were increased without changing the net charge of the residue for the pairs of atoms Nε, Hε, and Nδ, Hδ, respectively. The boundary between solute and solvent was described using one set of thin low dielectric spheres with a radius of 1.6 Å. A second set of thin spheres with a radius of 0.4 Å was used to calibrate ligand desolvation penalties. Scoring grids were precalculated using QNIFFT[77] for Poisson−Boltzmann electrostatic energies, SOLVMAP for ligand desolvation energies[78], and CHEMGRID for AMBER van der Waals energies[79]. Property-matched, property-perturbed, and property-extrema decoys of GPR139 ligands were generated using in-house scripts[80]. The control sets obtained were used to assess the docking grid's performance by measuring the enrichment of ligands over decoys. The final grid parameter selection was based on both the ligand enrichment and the accuracy of the predicted binding pose of **JNJ-63533054**.

The lead-like subset of the ZINC15 database[50] (cLogP ≤ 3.5 and molecular weight ≤ 350 Da, accessed November 2020) was docked to the GPR139 binding site using DOCK3.7 to screen for GPR139 ligands. The library contained over 235 million commercially available molecules, with successful docking achieved for 228 million of these. On average, each compound was evaluated in 3933 different orientations. Furthermore, for each orientation, approximately 178 conformations were scored. For every ligand successfully docked, the highest-scoring pose underwent optimization using a simplex rigid-body minimizer. To minimize the likelihood of encountering false positives, the top-scoring molecules were screened using a PAINS filter[81]. Compounds with high similarity ($T_c > 0.5$) to GPR139 ligands in the ChEMBL database were also excluded. The top 300,000 scoring molecules were clustered using Morgan fingerprints (radius = 2)[82] ($T_c > 0.5$) to enhance the diversity of the highest-ranked compounds. From each cluster, the molecule with the best score was chosen as the representative. Among the 13106 clusters formed, the top 1500 cluster representatives were visually inspected in compound selection.

**Preparation of chemical libraries.** In the hit optimization step, SMARTS patterns were constructed with the structural information and an SMARTS visualizer website (https://smarts.plus)[83]. Molecular pattern exploration was performed using OEToolkits from OpenEye software[84] and the Enamine's REAL space library (34 billion compounds) to search for analogs of hits from the virtual screening. For docking preparation, molecules from the SMILES were converted to the DB2 format using DOCK3.7 protocols. The program Omega[85] from OpenEye was used to generate conformational ensembles with a maximum of 2000 conformations for each rigid segment and a 0.25 Å RMSD diversity threshold between conformations.

**AlphaFold3 predictions.** AF3 models of the GPCR-ligand complexes were generated using version 3.0.0, installed on a GPU cluster with default parameters. Protein complexes were obtained using amino acid sequences from FASTA files, and ligand SMILES were extracted from the PDB. We used two random seeds (10 and 43) and produced 10 models per receptor-ligand complex. In each case, the best AF3 model based on the ranking score was used in the analysis. AF3 and experimental receptor structures were aligned using PyMOL, and the ligand heavy-atom RMSD was then calculated with Maestro software from Schrödinger LLC[86]. The number of receptor-ligand contacts was determined by counting receptor heavy atoms within 4 Å of any ligand heavy atom[87].

**Compound synthesis.** Compounds were synthesized by Enamine (https://enamine.net/) and Wuxi AppTec (https://www.wuxiapptec.com/). The purities of the active molecules were at least 90% and typically exceeded 95% (Supplementary Table S1). The key compounds that were resynthesized in-house had purities greater than 95%. Detailed synthetic procedures and analytical data are provided in the Supplementary Methods.

### In vitro biological assays

**Cell culture.** Primary screening and pharmacological characterization of compounds were performed using a CHO-K1 cell line that stably expresses human GPR139 (GPR139-CHOk1). The GPR139-CHOk1 cell line was kindly provided by H. Lundbeck A/S, Denmark[24]. GPR139-CHOk1 cells were grown in Ham's F-12 Kaighn's Modification Medium (ThermoFisher #21127030) supplemented with 10% dialyzed fetal bovine serum (US origin, ThermoFisher #26400044), 100 units/mL Penicillin-Streptomycin (ThermoFisher #15140122), and 1 mg/mL Geneticin (ThermoFisher #11811031). We counter screened against the human muscarinic acetylcholine $M_1$ receptor (M1R) stably expressed in CHO-K1 cells (M1R-CHOk1) to validate the specificity of our compounds induced responses for GPR139. M1R-CHOk1 cells were purchased from The Missouri S&T cDNA Resource Center (#CEM100TN00). M1R-CHOk1 cells were grown in Ham's F12 Medium (ThermoFisher #21765029) supplemented with 10% fetal bovine serum (Brazilian origin, ThermoFisher #10270106), 100 units/mL Penicillin-Streptomycin, and 0.25 mg/mL Geneticin. HEK293A cells transiently transfected with GPR139 were used for additional pharmacological assessment of signaling pathways activated by GPR139. HEK239A cells were generously provided by Dr. Asuka Inoue, Tohoku University, Japan. HEK293A cells were grown in Dulbecco's Modified Eagle Medium (ThermoFisher #31966047) supplemented with 10% dialyzed fetal bovine serum and 100 units/mL Penicillin-Streptomycin. Cell lines were maintained at 37 °C in humidified incubator equilibrated with 5% $CO_2$.

**Calcium mobilization assay.** 30,000 GPR139- or M1R-CHOk1 cells/well were seeded in black clear bottom tissue culture treated 96-well plates (Greiner #655090) and grown for 20–24 h. Cells were washed with HEPES buffer (Hank's Balanced Salt Solution (HBBS) supplemented with 20 mM HEPES, 1 mM $MgCl_2$, and 1 mM $CaCl_2$; pH 7.4). Cells were loaded with 50 μL of 2 μM Fluo-4 AM dye (ThermoFisher #F14202) in HEPES buffer supplemented with 2.5 mM probenecid and 0.01% Pluronic F-68 (ThermoFisher #24040032) for 1 h at 37 °C. Cells were washed with HEPES buffer. 50 μL HEPES buffer supplemented with 2.5 mM probenecid was added, and the cells were incubated for 10 min at 37 °C before measurement. 50 μL test compounds diluted in HEPES buffer from DMSO stocks (final DMSO concentration of maximally 0.3%) were added automatically after baseline measurements. The intracellular calcium mobilization was measured on a FlexStation 3 Multimode Microplate Reader (Molecular Devices) at 37 °C with filters for excitation of 485 nm and emission of 525 nm.

Our primary compound screening protocol consisted of two subsequent steps that allowed to screen for agonists and antagonists simultaneously. In the first step, the screening compounds were applied at 10 μM to test for agonist activity (agonist mode). Following 20 min incubation with the compounds added in the first step, in the

second step, an $EC_{80}$ concentration (800 nM) of the reference agonist **Lundbeck Cmp 1a** was applied to test for antagonist activity (antagonist mode). In the agonist mode (first step) response is seen only with compounds having agonist activity (**Cmp 1a**), but not with non-receptor-binding compounds (**Vehicle**) or antagonists (**NCRW0105-E06**) (Supplementary Fig. 14). In the antagonist mode (second step) if the applied screening compound in first step did not bind to the receptor, then the addition of **Lundbeck Cmp 1a** produces a response (**Vehicle + Cmp 1a**). However, if the screening compound has antagonist activity it can inhibit the response of **Lundbeck Cmp 1a** (**NCRW0105-E06 + Cmp 1a**), or if the screening compound has agonist activity and already strongly activated the receptor in the first step, a second addition of agonist does not activate the receptor again due to desensitization (**Cmp 1a + Cmp 1a**) (Supplementary Fig. 14). Hence, compounds with antagonist activity could be identified by having absence of response in the agonist mode (first step) and absent or significantly decreased $EC_{80}$ response of the reference agonist **Lundbeck Cmp 1a** in the antagonist mode (second step). No antagonist compounds were identified in this study.

HEK293A cells were reverse-transfected with human GPR139 for $Ca^{2+}$ mobilization experiments with transiently expressed GPR139. The construct has been generated and described previously by Nøhr et al. [25]. DNA and Lipofectamine 2000 Transfection Reagent (ThermoFisher #11668019) were separately diluted in Opti-MEM I Reduced Serum Medium (ThermoFisher #31985070). 10 ng GPR139 DNA and 250 nL Lipofectamine were used for each well. The dilutions were incubated at room temperature for 5 min. Then, DNA and Lipofectamine dilutions were combined in 1:1 volumetric ratio and incubated at room temperature for 20 min to obtain the transfection solutions. 10 μL transfection solution and 20,000 cells in 100 μL growth medium were distributed in each well of black clear bottom tissue culture treated 96-well plates that were previously coated with poly-D-lysine (Sigma-Aldrich #P0899). Cells were grown for approximately 48 h. A calcium mobilization assay was performed as described.

**Inositol monophosphate ($IP_1$) accumulation assay.** HTRF IP-One Gq kit (Cisbio #62IPAPEC) was used to detect the accumulation of inositol monophosphate ($IP_1$) as described in the work by Shehata et al.[26]. GPR139-CHOk1 cells were dissociated and resuspended in HEPES buffer. 50,000 cells in 5 μL and 5 μL test compounds diluted from DMSO stocks (final DMSO concentration of maximally 0.3%) in HEPES buffer supplemented with 40 mM LiCl were mixed in each well of white 384-well OptiPlate (PerkinElmer #6007290). The cells were incubated for 1 h at 37 °C. Then, 10 μL detection reagent composed of 2.5% Eu3 + -anti-IP1 antibody and 2.5% IP1-d2 diluted in lysis buffer was added, and the cells were incubated for 1 h at room temperature before measurements. Measurements were performed on EnVision Multimode Plate Reader (PerkinElmer) using excitation light of 340 nm and emission light of 615 nm and 665 nm. The time resolved-fluorescence resonance energy transfer (TR-FRET) 665 nm/615 nm ratio, which is inversely proportional to the $IP_1$ accumulation, was used to determine the $IP_1$ response.

**Bioluminescence resonance energy transfer (BRET) G protein activation assay.** The G protein dissociation assay developed by the lab of Kirill A. Martemyanov was used to measure the coupling of multiple G proteins by GPR139[57]. The assay was performed as described previously with minor modifications[49]. HEK293A cells were reverse transfected 20–24 h prior to assay. 50,000 cells were plated in each well of white opaque tissue culture treated 96-well plates (PerkinElmer #6005680) previously coated with poly-D-lysine. Transfection solutions were prepared as described, and 10 μL transfection solution was mixed with 100 μL cells in each well. The cells in each well were transfected with 18.9 ng GPR139, 6.3 ng Venus 1-155 Gγ2, 6.3 ng Venus 159-239 Gβ1, 6.3 ng masGRK3ct-NanoLuc and 12.6 ng Gα$_{i2}$, 12.6 ng Gα$_{i3}$,

12.6 ng Gα$_{oA}$, 12.6 ng Gα$_{q}$ or 25.2 ng Gα$_{12}$ using 375 nL Lipofectamine 2000 Transfection Reagent. Transfections with Gα$_{q}$ and Gα$_{12}$ contained 6.3 ng DNA of the active subunit S1 of pertussis toxin (PTX-S1). Each transfection was normalized to 150 ng total DNA with pcDNA3.1. The BRET biosensors were a kind gift from the Martemyanov lab. On the day of the experiment the cells were washed with HEPES buffer. Test compounds were diluted in HEPES buffer from DMSO stocks (final DMSO concentration of maximally 0.3%). NanoLuc Luciferase substrate NanoGlo−furimazine (Promega #N1120) was diluted 1000-fold (final) in HEPES buffer. NanoGlo and test compounds were added, and measurements were performed after 10 s for G$_{i3}$, 15 s for G$_{i2}$ and G$_{oA}$, 20 s for G$_{q}$, and 90 s for G$_{12}$−the timepoint where maximal signal occurred with each G protein, ensuring proper determination of the potency for the G protein coupling. Simultaneous dual emission was measured at 535 nm and 475 nm on LUMIstar Omega Microplate Reader (BMG Labtech) at room temperature. The BRET ratio of 535 nm/475 nm (Venus/NanoLuc), reflecting the dissociation of the G protein heterotrimer, was used to determine the G protein activation. The basal BRET ratio obtained with buffer condition was subtracted from the absolute BRET ratio at a given test compound concentration, and the responses were normalized to the one obtained with 10 μM **JNJ-63533054** for the activation of G$_{i2}$.

**Arrestin-3 recruitment assay.** The enhanced bystander BRET arrestin recruitment assay developed by the work of Donthamsetti et al.[58] was used to measure the arrestin-3 recruitment by GPR139. HEK293A cells were reverse transfected 20–24 h prior to the assay. 50,000 cells were plated in each well of white opaque tissue culture treated 96-well plates (PerkinElmer #6005680) previously coated with poly-D-lysine. The transfection solutions were prepared as described, and 10 μL transfection solution was mixed with 100 μL cells in each well. The cells in each well were transfected with 6 ng GPR139, 0.6 ng RLuc8-Arrestin3-Sp1, 6 ng GRK5, and 62.5 ng MEMlinker[doubly palmitoylated fragment of GAP43]-Citrine-SH3 using 200 nL Lipofectamine 2000 Transfection Reagent. Each transfection was normalized to 80 ng total DNA with pcDNA3.1. On the day of the experiment, the cells were washed with HEPES buffer and incubated for 2 h in HEPES buffer prior to measurements. Test compounds were diluted in HEPES buffer from DMSO stocks (final DMSO concentration of maximally 0.3%). *Renilla* luciferase 8 (RLuc8) substrate Coelenterazine h (Cayman Chemical #16894) was diluted in HEPES buffer to 5 μM (final). Coelenterazine h was added, and the cells were incubated for 5 min followed by stimulation with test compounds. Simultaneous dual emission was measured at 535 nm and 475 nm on LUMIstar Omega Microplate Reader (BMG Labtech) at 37 °C over 30 min. The BRET ratio of 535 nm/475 nm (Citrine/RLuc8), reflecting the translocation of arrestin from the cytosol to the membrane, was used to determine the arrestin recruitment. The BRET response-time course area under the curve of the basal response obtained with buffer condition was subtracted from the absolute response at a given test compound concentration, and the responses were normalized to the one obtained with 10 μM **JNJ-63533054**.

**Data analysis.** Pharmacological data and statistical analyses were performed in GraphPad Prism v9–10 (Graphpad Software Inc., San Diego, CA), which are also specifically described in the figure and table legends. The functional assay responses were normalized to buffer (0%) and **Lundbeck Cmp 1a** or **JNJ-63533054** (100%), for $Ca^{2+}$ and $IP_1$ experiments in the GPR139-CHOk1 stable cell line or $Ca^{2+}$, G protein activation, and arrestin-3 recruitment experiments in the HEK293A cells transiently transfected with GPR139, respectively. Data are from at least three independent experiments, unless stated otherwise. Concentration-response curves were fitted applying non-linear regression using the log (agonist) versus response model with variable slope. ANOVA was applied for comparing results and determining

statistical significance. Comparison of responses to the reference was performed using one-way ANOVA with Dunnett's multiple comparisons test at significance threshold of $p < 0.05$. Comparison of responses across assays and ligands was performed using two-way ANOVA with Tukey's multiple comparisons test at a significance threshold of $p < 0.05$.

### Structural biology experiments

**Protein engineering and expression of GPR139.** The GPR139 construct utilized in this research is a modified variant that was previously detailed for the structure of GPR139 bound with **JNJ-63533054**[49]. In brief, the C-terminal residues 322–353 were truncated, and the Xylanase fusion protein (with a molecular weight of 19.1 kDa corresponding to PDB entry 2B45) was attached to the N-terminus of GPR139. Additionally, the $S62^{2X39}V$ mutation was introduced to increase the stability of the receptor. Subsequently, the modified construct was cloned into the pFastBac1 vector with N-terminal hemagglutinin (HA) signal peptide, a Flag tag, and a deca-histidine (10× His) tag, followed by a site sensitive to tobacco etch virus (TEV) protease cleavage site. Protein expression was carried out using the Bac-to-Bac system (provided by Invitrogen) in *Spodoptera frugiperda* Sf9 insect cells. These cells were cultured at 27 °C for 48 h, and the cellular membrane was washed extensively in hypotonic buffer. The purification procedure of GPR139 is similar to that of a previous study[49], with the exception that the ligand (a racemic mixture of compound **1.1**) was used at a concentration of 50 μM throughout the process.

**GPR139-miniG$_{s/q}$-Nb35 complex expression, formation, and purification.** MiniGα$_{s/q}$ protein was expressed in the *E. coli* BL21 strain and induced by IPTG for 20 h at 20 °C. The resulting cell pellets were resuspended in buffer 1 containing 40 mM HEPES pH 7.5, 100 mM NaCl, 5 mM imidazole, 50 μM GDP, and 5 mM $MgCl_2$ along with 100 μM DTT and EDTA-free protease inhibitors. The cells were then sonicated, and the resulting supernatant was collected after centrifugation. The supernatant was incubated with Ni-NTA resin at 4 °C for 3 h. The resin was then washed with 30 CV wash 1 containing 20 mM HEPES pH 7.5, 250 mM NaCl, 10 mM imidazole, 50 μM GDP, 5 mM $MgCl_2$, and 10% glycerol. This was followed by 15 CV wash 2, where the imidazole concentration was increased to 30 mM, and the $MgCl_2$ concentration was decreased to 1 mM. The target protein was eluted with 7 CV elution using wash 2 as a basis; the imidazole concentration was increased to 500 mM, and the sodium chloride concentration was decreased to 50 mM. The eluted protein was further purified by size-exclusion chromatography using a Superdex 75 column (GE Healthcare). The resulting purified miniGα$_{s/q}$ protein was concentrated to 4 mg/ml and stored at −80 °C for further use in complex formation.

The Gβ1γ2 subunits were cloned into the pFastBac Dual vector and expressed in Sf9 insect cells. To streamline the purification process, a His-tag was fused to the N-terminus of the Gγ2 subunit, incorporating a TEV protease cleavage site for later removal. After 48 h, the cells were harvested and lysed in a hypotonic buffer. The buffer composition included 50 mM HEPES pH 7.5, 100 mM NaCl, protease inhibitor cocktail, 5 mM imidazole, 5 mM $MgCl_2$, and 1 mM DTT. The cell lysate was then centrifuged, and the supernatant was subjected to Ni-NTA affinity chromatography to isolate the His-tagged proteins. Following the initial purification, the fractions containing the Gβ1γ2 subunits were further purified by size-exclusion chromatography on a Superdex 75 column. The monomeric fractions containing Gβ1γ2 were pooled. The fraction of Gβ1γ2, concentrated to 3 mg/mL, was prepared for assembly with miniG$_{s/q}$ into heterotrimers.

To form the GPR139-miniG$_{s/q}$-Nb35 complex, we initiated the reaction by combining purified receptors with an excess molar ratio of G-protein. This mixture was then incubated at 24 °C for 1 h to facilitate coupling. After the addition of the apyrase, the reaction mixture was incubated at 25 °C for an additional 2 h. Then, a 1.2 molar excess of

Nb35 was introduced into the reaction and allowed to incubate with the forming complex at 4 °C overnight.

The purification of the GPR139-miniG$_{s/q}$-Nb35 complex was accomplished by size exclusion chromatography using a Superdex 200 10/300 GL column with the buffer of 20 mM HEPES (pH 7.5), 100 mM NaCl, 0.00075% w/v LMNG, 0.00025% GDN, 0.0001% w/v cholesterol hemisuccinate (CHS), 100 mM TCEP, and 10 μM ligand. Finally, the peak fractions containing the purified complex were collected and concentrated to a final concentration of 3.5 mg/mL for cryo-EM studies (Supplementary Fig. 8).

**Cryo-EM sample preparation and image acquisition.** A 3 μL aliquot of the purified GPR139-miniG$_{s/q}$-Nb35 complex sample was placed onto a glow-discharged holey carbon grid (CryoMatrix Amorphous alloy film R1.2/1.3, 400 mesh). Vitrification was carried out using a Vitrobot Mark IV (Thermo Fisher Scientific), with the instrument's chamber conditions set to 100% humidity and 4 °C. The sample underwent blotting for 2.5 s at a blot force setting of 2. Cryo-electron microscopy images were acquired with a Krios G4 microscope operating at 300 kV, equipped with a Falcon 4 Summit direct electron detector and a Quantum energy filter (Thermo Fisher Scientific). The data were collected in EFTEM nanoprobe mode, employing a 50 μm C2 aperture and a calibrated magnification of 130,000x with pixel size of 0.96 Å. Movies were recorded in the EER format, with each frame subjected to an accumulated electron dose of 60 electrons per $Å^2$. The EPU software (Thermo Fisher Scientific, v.1.11.0) facilitated automated data acquisition within a defocus range from −1.2 to −2.0 μm (Supplementary Fig. 8c, d). Detailed information is included in Supplementary Table 8.

**Cryo-EM data processing and 3D reconstruction.** For the GPR139-miniG$_{s/q}$-Nb35 complex, 13,374 movies were collected and analyzed by cryoSPARC v4[88]. Patch motion correction was applied for beam-induced motion, and contrast transfer function (CTF) parameters were determined for each dose-weighted micrograph using cryoSPARC's patch CTF estimation. From the autopicked 5,902,555 particles[89], three rounds of 2D classification were conducted, resulting in 1,784,244 particles for initial model construction and 3D classification. The optimal class, with its 138,265 particle projections, underwent homogenous and non-uniform refinement. The final density map was resolved to 3.2 Å per gold standard Fourier shell correlation (FSC) at the 0.143 criterion (Supplementary Fig. 8e). Local resolution was gauged using cryoSPARC's Local Resolution Estimation, and automatic local sharpening of the EM density maps was done with DeepEMhancer (Supplementary Fig. 8f)[90].

**Model building and refinement.** To model the GPR139-Compound **1.1(S)**-miniG$_{s/q}$-Nb35 structure, the previously solved GPR139-**JNJ-63533054**-miniG$_{s/q}$-Nb35 structure (PDB: 7VUH) served as the initial model[49]. The cryo-EM structural model was aligned to the electron microscopy density using Chimera[91], with manual adjustments in Coot[92], and refinement in PHENIX via phenix.real_space_refine[93]. The model's accuracy was checked with MolProbity. Structural figures were created with Chimera and PyMOL (http://www.pymol.org). Refinement quality is detailed in Supplementary Table 8, and model fitting was assessed by refining against a half-map and comparing FSC curves of map versus model between half-maps and the full model.

### In vitro ADME assays

**Kinetic solubility.** Kinetic solubility, utilizing test compound from 10 mM DMSO stock solution, was measured at a final compound concentration of 100 μM and 1% DMSO. Test compound was added to 100 mM potassium phosphate buffer and incubated at 37 °C for at least 20 h in a heater-shaker. After incubation, the samples were centrifuged at $3000 \times g$ at 37 °C for 30 min to pellet insoluble material and

an aliquot of the supernatant was taken for analysis. After dilution of the sample, the concentration of dissolved compound was quantified by liquid chromatography coupled to triple quadrupole mass spectrometry (LC-MS/MS).

**Metabolic stability in the presence of mouse liver microsomes.** Metabolic stability was determined in 0.5 mg/ml mouse liver microsomes at a compound concentration of 1 μM in 100 mM $KPO_4$ buffer, pH 7.4, in a total incubation volume of 500 μl. The reaction was initiated by addition of 1 mM NADPH. At various incubation times, i.e., at 0, 5, 10, 20, 40, and 60 min, a sample was withdrawn from the incubation, and the reaction was terminated by addition of cold acetonitrile with warfarin as an internal standard. The amount of parent compound remaining was analyzed by LC-MS/MS[70].

### In vivo behavioural assays

**Pharmacological treatments for animals.** Adult male C57BL/6 J mice (3-6 months old) were provided by Charles River. Both females and males (15 females and 9 males) were used in this study. Sex was evenly distributed across experimental groups. Sex was not considered in the study design or analysis, as no reports indicate differential expression levels of GPR139 between male and female mice. They were housed with ad libitum food and water, in a temperature and humidity-controlled environment under a standard 12 h light/dark cycle. The experiments were performed in agreement with the European Council Directive (86/609/EEC) and approved by the local Animal Ethics Committee at Karolinska Institute (3218-2022). All efforts were made to minimize suffering and the number of animals used. Mice were treated with either 10% DMSO + 10% solutol HS-15 + 80% saline (vehicle, $n = 8$), compound **1.5** ($n = 8$, 30 mg/kg i.p.), or **JNJ-63533054** ($n = 8$, 30 mg/kg i.p.), both of which were dissolved in 10% DMSO + 10% solutol HS-15 + 80% saline. Mice were placed in an open field arena 10 min after the injection and recorded for 50 min. Total distance travelled and the time spent in the periphery arena were measured using an automated video tracking system (EthoVision XT 11.5; Noldus). One-way ANOVA followed by Dunnett's multiple comparisons test and Two-way ANOVA followed by Tukey's multiple comparisons test were used. Brain and blood samples were taken immediately after the locomotion test. The brain samples were quickly frozen, and blood was drawn from the cervical vein upon decapitation into EP tubes. The samples were immediately centrifuged for 10 min at $3000 \times g$ at 4 °C, and the plasma was transferred into clean polypropylene tubes. All samples were stored at −80 °C before analysis.

### Bioanalysis of compounds in mouse plasma and brain

**Chemicals.** Warfarin (Cat. No. A2250) was purchased from Sigma-Aldrich Sweden AB (Sweden). Acetonitrile (LiChrosolv®, liquid chromatography grade) was purchased from Merck AB (Sweden). Formic acid (HiPerSolv Chromanorm®) and methanol (MeOH; LiChrosolv®, mass spectrometry grade) were purchased from VWR International AB (Sweden). MilliQ water was obtained from a Milli-Q® system (18.2 mΩ cm at 25 °C, Millipore, Billerica, MA, USA).

**Bioanalysis.** **1.5(S)** and **JNJ-63533054**, and the internal standard warfarin, were quantified in plasma and brain tissue samples, using ultra-performance liquid chromatography-tandem mass spectrometry (UPLC-MS/MS). Samples, respective blanks, standards, and quality controls (QCs) were included in each bioanalytical run. For both compounds, the calibration curves in plasma included standards at 0.5–10 μM, and QCs at 1.9–7.6 μM, and those in brain tissue homogenate included standards at 1–25 μM, and QCs at 1–20 μM.

**Sample preparation.** Plasma samples were first diluted 1:2 (v:v) in blank plasma on a 96-well plate (Corning). Thereafter, the samples,

along with blank plasma, standards, and QCs, were precipitated by adding 180 μL ice-cold acetonitrile containing 100 nM warfarin to 20 μL sample. Samples were vortexed and centrifuged at $2465 \times g$, at 4 °C, for 20 min (Centrifuge 5810 R, Eppendorf, Germany). The supernatants were transferred to a new 96-well plate and 2 μL were injected onto the UPLC-column. One hemisphere of the brains was homogenized 1:4 (w:v) with 0.1 % formic acid in 1:1 acetonitrile and methanol (v:v) in tubes containing ceramic beads (Precellys® Hard Tissue Homogenizing CK28 (inert 2.8 ceramic, zirconium oxide, beads), 2 mL, polypropylene co-polymer, Bertin Technologies). The mechanical homogenization was performed using a Minilys homogenizer (Bertin Technologies, France) for 30 s at the highest speed (5000 rpm). The tubes were centrifuged at $5204 \times g$ for 7 min, and the supernatant was transferred to an Eppendorf tube. Subsequently, brain homogenate supernatants, along with supernatants from blank brain homogenate, standards, and QCs, were treated as described for diluted plasma samples.

**Quantification of compounds.** The bioanalysis system consisted of an Acquity UPLC system (Waters Corporation, Milford, MA, USA) coupled to a triple quadrupole mass spectrometer (XevoTM TQ-S micro, Waters Corporation, Ireland). A linear elution gradient with an initial gradient of 100% Mobile phase A (MPA) was applied and maintained until 0.5 min. Thereafter, mobile phase B (MPB) was increased to 100% until 1.3 min, and maintained until 1.8 min. Between 1.8 and 2 min, MP B was decreased to 5%. MPA consisted of 0.1% formic acid in water, and MP B of 0.1% formic acid in acetonitrile. The flow rate was set to 0.5 mL/min. Chromatographic separation of **1.5(S)**, **JNJ-63533054**, and warfarin was performed on a BEH C8 column (2.1 × 50 mm, 1.7 μm, Waters, kept at 60 °C). Electrospray ionization source was operated in positive mode set to a capillary voltage of 3 kV, a desolvation gas temperature of 500 °C, and a desolvation gas flow of 1100 L/h. Multiple reaction monitoring transitions monitored were 313.3 > 105.4, 317.1 > 196, and 309.2 > 163.0 for **1.5(S)**, **JNJ-63533054**, and warfarin, respectively. Quantification was performed using TargetLynx software (ver. 4.2, Waters). Estimated concentrations in plasma and brain homogenate samples were multiplied by three and four, respectively, to obtain the concentration in undiluted plasma and brain tissue. Outliers ($N = 2$, $n = 3$) were identified using the ROUT method for identification of outliers in GraphPad Prism (GraphPad Prism version 10.0.0 for Windows, GraphPad Software, Boston, MA, USA).

### Ethics statement

In vivo experiments using adult male mice were performed in agreement with the European Council Directive (86/609/EEC) and approved by the local Animal Ethics Committee at Karolinska Institute (3218-2022). All efforts were made to minimize suffering and the number of animals used.

### Reporting summary

Further information on research design is available in the Nature Portfolio Reporting Summary linked to this article.

## Data availability

The ZINC15 library is available at https://zinc15.docking.org. The GPR139 cryo-EM structure used in the molecular docking calculations and predicted complexes with compounds **1**, **2**, **3**, **4**, **5**, and **1.1** are provided as Supplementary Data 1. The compounds tested are listed in the Supplementary Information and Source Data File. The synthetic methods, chemical identities, purities (LC/MS), and spectroscopic analysis ($^1$H and $^{13}$C NMR) for key compounds in this study are provided in the Supplementary Information. The cryo-EM data generated in this study have been deposited in the PDB database under accession code 9M42. Source data are provided with this paper.

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

## Acknowledgements

J.C. received funding from the Olle Engkvist Foundation, Swedish Brain Foundation, and the Swedish strategic research program eSSENCE. This research was partially supported by the project AI4Research at Uppsala University. I.C. was funded by a postdoctoral fellowship provided by the Sven och Lilly Lawski foundation. The computations were enabled using resources provided by the National Academic Infrastructure for Supercomputing in Sweden (NAISS) (partially funded by the Swedish Research Council through grant agreement no. 2022-06725) and the supercomputing resource Berzelius provided by National Supercomputer Centre at Linköping University and the Knut and Alice Wallenberg Foundation. This project was supported by the Compound Center at the Chemical Biology Consortium Sweden (CBCS). CBCS is funded by the Swedish Research Council (2021-00179) and SciLifeLab. This study made use of the NMR Uppsala infrastructure, which is funded by the Department of Chemistry–BMC and the Disciplinary Domain of Medicine and Pharmacy. The authors thank Pawel Baranczewski and the Uppsala University Drug Optimization and Pharmaceutical Profiling Platform (UDOPP) for determination of in vitro pharmacokinetic properties. The authors thank OpenEye Scientific Software for the use of OEChem and OMEGA at no cost. B.T. thanks the Faculty of Health and Medical Sciences, University of Copenhagen, for funding this project. J.C. and A.S.H. are part of the European Cooperation in Science and Technology (COST) action CA18133 (ERNEST).

## Author contributions

I.C., B.T., H.B-O, A.S.H. and J.C. conceived the project. I.C. performed the molecular docking calculations and analyzed the results under the supervision of J.C. B.T. designed and performed the pharmacological experiments and data analysis under the supervision of A.S.H. and H.B-O. L.S. and Xu.Z. obtained and refined the cryo-EM structure under the supervision of Z-J.L. M.P. performed pharmacological experiments under the supervision of B.T. Xi.Z. and Y.Y. performed in vivo behavioural experiments under the supervision of P.S. D.D.V. synthesized compounds under the supervision of J.K. A.V.T., D.S.R. and Y.S.M. supervised synthesis of compounds at Enamine and provided analytical data. F.B. and A.S. performed the analysis of mouse plasma and brain tissue. I.C., B.T., A.S.H. and J.C. wrote the manuscript with input from all co-authors. All co-authors discussed the results and revised the manuscript.

## Funding

## Competing interests

J.C. is a founder of DareMe Drug Discovery Consulting. Y.S.M. is a VP of Sales and Marketing at Enamine Ltd. and a scientific advisor at Chemspace LLC. A.V.T. and D.S.R. are employees of Enamine, Ltd. The remaining authors declare no competing interests.
