## [Transparent Peer Review file · Nature Communications]

Ultra-large virtual screening unveils potent agonists of the neuromodulatory orphan receptor GPR139

Corresponding Author: Professor Jens Carlsson

Version 0:

Reviewer comments:

Reviewer #1

(Remarks to the Author)

In their manuscript, Ultra-large virtual screening unveils potent agonists of the neuromodulatory orphan receptor GPR139', the authors docked 235 million compounds into the orthosteric ligand binding site of GPR139, an 'orphan' receptor involved in movement, motivation and reward. More recently GPR139 has also been linked to opioid and dopamine neuromodulatory systems, making it a promising drug target for the treatment of diseases like Parkinson's disease as well as neuropsychiatric, motor and metabolic disorders. Based on the docking study and structure-guided hit optimization, they were able to identify an agonist for the receptor that simulate Ca²⁺ mobilization and IP1 accumulation with higher potency compared to previously known reference GPR agonists. Furthermore, using BRET-based signaling assays, the authors showed that the S-enantiomer of the compound 1.1 efficiently activates G proteins of the Gi family as well as the G proteins Gq and G12. The binding pose of the developed compound 1.1 was then evaluated by cryo-EM structure determination of GPR139 in complex with the ligand and miniGs/q. The structure was obtained at a nominal resolution of 3.2 Å, allowing them to build a receptor model including side chains and the bound ligand in the orthosteric binding pocket. Interestingly, the authors showed that the ligand binding pose of compound 1.1 was reasonably well predicted using AlphaFold3. However, for other orphan receptors, AF3 failed to identify the correct binding site, suggesting that experimental structures, especially when they are bound to available tool compounds, still provide the best template structures to identify suitable pockets for docking studies. Finally, the authors tested their compound in mice to assess its activity on animal behavior in an open-field test. While the most potent ligand showed low metabolic stability, the derivative 1.5 resulted in a reduction in locomotor activity and increased anxiety-like behavior.

Despite the identified hit compounds showing significant structural similarity to previously developed compounds (Cmp 1a, JNJ-63533054, and TAK-041), and the relatively low in vivo value of compound 1.1 due to its poor metabolic stability, this study provides very valuable SAR data and new structural insights into ligand recognition of the GPR139. This makes the work highly relevant to both ligand discovery and the broader GPCR field. Furthermore, while some earlier studies (Matsuo et al., 2005) provided first hints for GPR139-dependent G12 coupling, this manuscript offers the first direct evidence, providing more convincing data for G12 activation. This provides novel insights into the biology of GPR139. Altogether, I believe this work could be suitable for publication in Nature Communications once the following points have been addressed:

- 1) The orientation of the chlorophenylring in Fig. 1a seems to be different compared to the published 7VUG structure, where the chloride points towards TMs 6/7. I assume that pose 2 are being displayed that have been described in Zhou et al., 2022. The authors should clarify this.
- 2) Based on the current organization of the manuscript, it is hard to understand, why the authors used the S-enantiomer of compound 1.1 for structure determination. This becomes clear when the authors provide data for the signaling signature of the agonist on page 17. The authors should consider moving the latter part in front of the structural description to allow the reader to better understand the impact of the stereochemistry on ligand efficacy.
- 3) Fig. 4d: Please, remove the 3.5Å above the ligands
- 4) The density for ICL3 (residues 215-223) and ECL1 (93-98) is very fragmented. These residues should not be modelled in the structure
- 5) SFig. 8: in addition to the half-map FSC, the authors should also provide the model vs. map FSC and provide the FSC = 0.5 criterion for the latter.
- 6) Page 14: The authors should provide the residues between they measured the distances between the position of TMs 1 and 7 in the new and JNJ-63533054-bound structure. It seems to me that TM1 does not move that much compared to TM7.

TM2 moves also by 2A outward, when measured between CA of D89.

7) Comparison of the ligand binding poses between JNJ-63533054 (PDB ID 7VUG) and compound 1.1 (the provided structure), the phenylring is shifted upwards by 2.3 Å, resulting also in a different position of the carbonyl oxygen. The authors should describe these differences and what might be causing them.

Reviewer #2

(Remarks to the Author)

The manuscript by Israel Cabeza de Vaca et al. describes discovery of novel agonists of poorly explored target GPR139 that has a potential to be used for the treatment of neurological disorders. The compounds have comparable potency with previously identified ligands but have novel scaffolds. The remarkable achievement is successful utilization of low 3.2 Å resolution CryoEM structure of the receptor complexed to a ligand in virtual screening. It allowed to identify ligands with 20% accuracy as 14 out of 68 hits from 200 million compounds library showed activation of the receptor at 10 μM concentration in calcium mobilization assay. Expansion of the screen to 36 billion compounds by extracting analogs of the hits from the first screen allowed to improve the potency several fold, but did not produce compounds with much improved activity. Identified compounds have been carefully characterized in in vitro assays and showed biological activity in mice. The manuscript brings significant new knowledge in the field of GPCRs targeting and establishes the utility of new approaches in drug discovery in general.

It has few minor drawbacks, however:

The phrase at the bottom of page 5 "The utility of experimentally determined and machine-learning generated receptor-ligand complex structures in virtual screening is also considered" is vague and should be replaced with a clearer statement.

Figure 3a is confusing. Looks like "R" has been used in two different ways. Would be better to have different designation for substituents and a preserved part of the agonist.

Concentration units are missing in Supplementary Tables 4 and 6.

Would be beneficial to discuss the reasons the screens did not identify compounds that are significantly more potent than previously published one: imperfections with utilized receptor structure, its flexibility, limited diversity of the virtual libraries or low cut-off of molecular weight of the libraries.

I would recommend publishing the manuscript after correcting these minor drawbacks.

Reviewer #3

(Remarks to the Author)

In the current manuscript, Cabeza de Vaca et al., present a novel strategy to advance the discovery of ligands for orphan receptors. They selected the orphan G protein-coupled receptor (GPCR) GPR139 as the receptor of interest. However, their approach can be applied to any orphan receptor with a known cryo-EM structure.

The authors combined computational docking to identify novel ligands, followed sequentially by cell-based approaches to pharmacologically profile these ligands. Ultimately, they tested the newly identified ligands in vivo to assess their ability to modulate GPR139 activity in the central nervous system (CNS).

The authors demonstrated that calcium mobilization is not the only signalling pathway activated by GPR139, as the receptor also couples with Gi2, Gi3, GoA, Gq, and G12 proteins, in addition to recruiting arrestin.

The data presented are solid and robust. However, a few additional controls could eliminate any remaining doubts regarding the specificity of the recorded signals:

1. Unspecific Binding of Compound 1.1

Prior to characterizing the stereoisomers of compound 1.1 in signalling assays, it would be beneficial to demonstrate that compound 1.1 does not bind non-specifically. This could be addressed by measuring calcium mobilization in cells treated with compound 1.1 in the absence of the receptor.

2. G12 Coupling and Expression Levels

The authors report that GPR139 activates G12 coupling. In their experimental setup, they used double the amount of Gα12 DNA (25.2 ng) compared to other isoforms (12.6 ng each for Gαi2, Gαi3, GαoA, and Gαq). This raises the question of whether the observed interaction with G12 results from a higher expression level. Considering also that the amount of receptor transfected in every case is the same amount (18,9 ng). Would decreasing the amount of Gα12 DNA abolish the interaction?

Additionally, to place these findings in a more physiological context, it would be helpful to discuss any known relevance of Gα12 signalling in the CNS.

3. Blood-Brain Barrier Penetration Claim

In their in vivo experiments, the authors state: "To study whether our compounds could penetrate the blood-brain and activate the GPR139 receptor". The authors show that the compound can modify locomotion and thigmotaxis as the reference compound (JNJ-63533054). Since the receptor is known to be expressed in the CNS it is plausible to hypothesize that the compound crosses the BBB however they do not visualize such a process nor they compared the effect of the agonist on GPR139 in KO animal.

Version 1:

Reviewer comments:

Reviewer #1

(Remarks to the Author)

The authors sufficiently addressed my concerns and I can now recommend publication of this work in Nature Communication.

Reviewer #2

(Remarks to the Author)

The authors have addressed all the concerns, and the manuscript can be published in its current form.

Reviewer #3

(Remarks to the Author)

I have carefully considered the authors' responses to the reviewers' comments and am pleased to confirm that all concerns have been thoroughly addressed.

I am satisfied with the revisions and have no further suggestions. I recommend that the manuscript be accepted for publication in its current form.

We have carefully considered all comments from the reviewers and performed additional analysis and experiments, leading to a substantially stronger manuscript. We have also modified several sections and figures in the revised manuscript based on the reviewers' comments. Please find below point-by-point responses to the referees with their comments in **Blue**. Relevant changes to the manuscript have been marked in **yellow** to facilitate the review process.

Reviewer #1

The first reviewer appreciated our study: *“this study provides very valuable SAR data and new structural insights into ligand recognition of the GPR139. This makes the work highly relevant to both ligand discovery and the broader GPCR field.”*. The reviewer supported publication: *“I believe this work could be suitable for publication in Nature Communications”*. We respond to the reviewer's questions below:

1. Reviewer: *“The orientation of the chlorophenylring in Fig. 1a seems to be different compared to the published 7VUG structure, where the chloride points towards TMs 6/7. I assume that pose 2 are being displayed that have been described in Zhou et al., 2022. The authors should clarify this.”*

We thank the reviewer for noticing this difference. The orientation of the chlorophenyl ring in Figure 1a indeed corresponds to “pose-2”, as described in Supplementary Figure S7b of Zhou *et al.* (<https://doi.org/10.1038/s41422-021-00591-w>). During the preparation of the virtual screen, we carefully examined the previously reported poses and selected “pose-2” because this complex provides a more plausible fit within the binding pocket based on visual inspection of receptor-ligand interactions and overall consistency with our structural data. We have clarified this point in the methods section of the manuscript (page 26).

2. Reviewer: *“Based on the current organization of the manuscript, it is hard to understand, why the authors used the S-enantiomer of compound 1.1 for structure determination. This becomes clear when the authors provide data for the signaling signature of the agonist on page 17. The authors should consider moving the latter part in front of the structural description to allow the reader to better understand the impact of the stereochemistry on ligand efficacy.”*

We are grateful for this comment from the reviewer as we realized that the manuscript text was not sufficiently clear. In the structure determination, a racemic mixture of compound **1.1** was used. The structural model was constructed based on the experimental EM density, with the stereoisomers of the ligand placed to best match the observed EM map. The S-enantiomer of compound **1.1** (**1.1(S)**) showed a clear and consistent fit to the ligand density. In contrast, when we attempted to dock compound **1.1(R)** into the same density, the fit was suboptimal and did not adequately account for the shape of the EM map. To further validate this assignment, we then tested the enantiopure compounds **1.1(S)** and **1.1(R)** using calcium mobilization assay. The experiments demonstrated that **1.1(S)** effectively activates GPR139, whereas **1.1(R)** exhibited markedly reduced agonist activity, as shown in Supplementary Figure S11. Together, structural and pharmacological data support the assignment of **1.1(S)** as the more active enantiomer in the reported structure. The organization of the manuscript hence reflects how the study was performed, and we have added this information on pages 14-15, 19, and 36.

3. Reviewer: *“Fig. 4d: Please, remove the 3.5Å above the ligands”*

We thank the reviewer for noticing this error. The “3.5 Å” label has been removed from Figure 4d (page 17).

4. Reviewer: *“The density for ICL3 (residues 215-223) and ECL1 (93-98) is very fragmented. These residues should not be modelled in the structure”*

We thank the reviewer for pointing this out. Due to the inherent flexibility of loop regions, the local resolution for ICL3 (residues 215–223) and ECL1 (residues 93–98) is indeed relatively low in the cryo-EM map. To ensure clarity and improve the overall accuracy of the structural model, we have deleted the residues 93–98 and 215–223 from the final deposited model.

5. Reviewer: *“SFig. 8: in addition to the half-map FSC, the authors should also provide the model vs. map FSC and provide the FSC = 0.5 criterion for the latter”*

We thank the reviewer for this suggestion. The FSC curve has been included in the revised version of the manuscript (Supplementary Figure S8).

6. Reviewer: *“The authors should provide the residues between they measured the distances between the position of TMs1 and 7 in the new and JNJ-63533054-bound structure. It seems to me that TM1 does not move that much compared to TM7. TM2 moves also by 2Å outward, when measured between CA of D89.”*

We agree with the reviewer that the description of the structural differences was not sufficiently clear. In this section, our intention was to highlight the transmembrane helices with the largest and smallest displacements. We measured the displacement distances between residues on TM1–TM7 in the superimposed new structure and the **JNJ-63533054**-bound structure. The C α displacement distances for each helix and the residue numbers have been included in the revised manuscript (TM1: F27–1.7 Å, TM2: D89–2.1 Å, TM3: K102–1.1 Å, TM4: L158–0.5 Å, TM5: V182–0.8 Å, TM6: Y250–1.6 Å, TM7: H264–1.6 Å). These measurements quantitatively reflect the relative movement of each transmembrane helix in the new structure compared to the **JNJ-63533054**-bound conformation. We have added this information to pages 15-16 of the revised manuscript.

7. Reviewer: *“Comparison of the ligand binding poses between JNJ-63533054 (PDB ID 7VUG) and compound 1.1 (the provided structure), the phenylring is shifted upwards by 2.3 Å, resulting also in a different position of the carbonyl oxygen. The authors should describe these differences and what might be causing them.”*

We thank the reviewer for this insightful comment. In the revised manuscript, we provide an analysis of the structural differences between the ligand binding modes of **JNJ-63533054** (PDB accession code: 7VUG) and compound **1.1** (pages 15-17 and Supplementary Figure S9).

Reviewer #2

The second reviewer was enthusiastic about our work: *“The manuscript brings significant new knowledge in the field of GPCRs targeting and establishes the utility of new approaches in drug discovery in general.”*. This reviewer also supported publication of the manuscript in Nature Communications: *“I would recommend publishing the manuscript after correcting these minor drawbacks.”*. We address the minor comments from the reviewer below:

1. Reviewer: *“The phrase at the bottom of page 5 “The utility of experimentally determined and machine-learning generated receptor-ligand complex structures in virtual screening is also considered” is vague and should be replaced with a clearer statement.”*

We thank the reviewer for this helpful suggestion. We agree that the original phrasing was vague and have revised the sentence to more clearly reflect our results (pages 5-6): *“We also assessed whether receptor-ligand complex structures generated by machine learning methods could serve as effective alternatives to experimentally determined structures for virtual screening targeting understudied GPCRs.”*

2. Reviewer: *“Figure 3a is confusing. Looks like “R” has been used in two different ways. Would be better to have different designation for substituents and a preserved part of the agonist.”*

We thank the reviewer for pointing out that Figure 3a was not sufficiently clear. In the revised manuscript, we have modified the representation of the substituents in the figure and extended the caption (page 13).

3. Reviewer: *“Concentration units are missing in Supplementary Tables 4 and 6.”*

We thank the reviewer for this comment. However, we have expressed potencies in Supplementary Tables S4 and S6 as pEC₅₀ values. Given that pEC₅₀ is a logarithmic transformation of concentration that by standard practice is expressed in molar (M), this quantity is dimensionless and unitless. Hence, concentration units are not missing and have not been added.

4. Reviewer: *“Would be beneficial to discuss the reasons the screens did not identified compounds that are significantly more potent than previously published one: imperfections with utilized receptor structure, its flexibility, limited diversity of the virtual libraries or low cut-off of molecular weight of the libraries.”*

We agree with the reviewer and have added a section addressing these important points in the discussion. As our most potent compound exhibits very high activity (EC₅₀ = 8 nM), identifying significantly more potent compounds may be challenging. Since we have already achieved a potency comparable to that of a clinical candidate (TAK-041), subsequent optimization efforts should increasingly focus on other important drug-like properties, e.g. pharmacokinetics and selectivity. As the reviewer points out, one drawback of relying on commercial chemical space is the limited size and diversity of these libraries. In our experience, there are generally too few analogs for each scaffold to successfully carry out lead optimization, and the synthesis of compounds beyond those available in commercial libraries is often required (e.g., see ref. 66, <https://www.nature.com/articles/s41467-025-56893-9>). Additionally, although our cryo-EM structure provides an excellent starting point for further optimization, accounting for receptor flexibility and using more accurate methods for predicting ligand affinity can likely improve compound design. These more general aspects of structure-based drug design are now included in the revised discussion of the manuscript (page 24).

Reviewer #3

The third reviewer was also positive about our work and noted that the “*data presented are solid and robust*”. The reviewer requested a few additional controls, and we address these questions below:

1. Reviewer: “*Unspecific Binding of Compound 1.1. Prior to characterizing the stereoisomers of compound 1.1 in signalling assays, it would be beneficial to demonstrate that compound 1.1 does not bind non-specifically. This could be addressed by measuring calcium mobilization in cells treated with compound 1.1 in the absence of the receptor.*”

We agree with the reviewer that controls for unspecific interactions are crucial. We have already performed such experiments for all our screening hits, including compound 1.1. In these experiments, we counter-screened at the human muscarinic acetylcholine M₁ receptor in the same assay and background cell line (M1R-CHO cells). As shown in Supplementary Figures S2 and S5, we demonstrated absence of response for all compounds, corroborating GPR139 specificity. We revised the manuscript text to improve clarity regarding the experiments assessing target specificity (pages 8 and 11)

2. Reviewer: “*G₁₂ Coupling and Expression Levels. The authors report that GPR139 activates G₁₂ coupling. In their experimental setup, they used double the amount of G_{α12} DNA (25.2 ng) compared to other isoforms (12.6 ng each for G_{αi2}, G_{αi3}, G_{αoA}, and G_{αq}). This raises the question of whether the observed interaction with G₁₂ results from a higher expression level. Considering also that the amount of receptor transfected in every case is the same amount (18,9 ng). Would decreasing the amount of G_{α12} DNA abolish the interaction? Additionally, to place these findings in a more physiological context, it would be helpful to discuss any known relevance of G_{α12} signalling in the CNS.*”

We thank the reviewer for these suggestions. The DNA amounts of transfected G proteins were based on the protocol described in the original publication by the team that developed the assay (<https://www.science.org/doi/10.1126/scisignal.aab4068>). The DNA amounts have been carefully optimized to ensure signal specificity, minimal background, and maximal response. Given that different constructs have different expression levels it is not surprising that different DNA amounts of each G protein are required to achieve a measurable signal. Therefore, we would argue that decreasing the amount of G₁₂ DNA does not provide further insights into the validity of the results, and the suggested experiment is redundant. It is possible that decreasing the amount of G₁₂ DNA would decrease the signal or even abolish it, but this would solely be due to decreased expression. Studies in native cells will be required to understand the relevance of GPR139 signaling via different G proteins. In the revised manuscript, we have added a discussion regarding G₁₂ signaling in the CNS to place our findings in a more physiological context, as suggested by the reviewer (page 26).

3. Reviewer: *“Blood-Brain Barrier Penetration Claim. In their in vivo experiments, the authors state: *‘‘To study whether our compounds could penetrate the blood-brain and activate the GPR139 receptor’.* The author show that the compound can modify locomotion and thigmotaxis as the reference compound (JNJ-63533054). Since the receptor is known to be expressed in the CNS it is plausible to hypothesis that the compound cross the BBB however they do not visualise such a process nor they compared the effect of the agonist on GPR139 in KO animal.”

To assess whether the compounds evaluated *in vivo* can penetrate the blood-brain barrier, we analyzed mouse plasma and brain tissue. The concentrations of **1.5(S)** and **JNJ-63533054** were determined one hour after administration of 30 mg/kg i.p., and these experiments clearly demonstrate that both compounds reach the brain. For **1.5(S)**, the total plasma concentration was determined to be 8.0 μM , and the total brain tissue concentration was 4.8 μM . For **JNJ-63533054**, the corresponding concentrations were 20.5 and 24.8 μM , respectively. Both tested compounds thus reached the brain and were present at concentrations exceeding their *in vitro* potencies. These results have been added to pages 22 and 41-44 of the revised manuscript.